# Control of transcription elongation and DNA repair by alarmone ppGpp

Jacob W. Weaver[1,4], Sergey Proshkin[2,4], Wenqian Duan[1,4], Vitaly Epshtein[1], Manjunath Gowder[1,3], Binod K. Bharati[1,3], Elena Afanaseva[2], Alexander Mironov[2], Alexander Serganov ®[1,5] ✉ & Evgeny Nudler ®[1,3,5] ✉

Second messenger (p)ppGpp (collectively guanosine tetraphosphate and guanosine pentaphosphate) mediates bacterial adaptation to nutritional stress by modulating transcription initiation. More recently, ppGpp has been implicated in coupling transcription and DNA repair; however, the mechanism of ppGpp engagement remained elusive. Here we present structural, biochemical and genetic evidence that ppGpp controls *Escherichia coli* RNA polymerase (RNAP) during elongation via a specific site that is nonfunctional during initiation. Structure-guided mutagenesis renders the elongation (but not initiation) complex unresponsive to ppGpp and increases bacterial sensitivity to genotoxic agents and ultraviolet radiation. Thus, ppGpp binds RNAP at sites with distinct functions in initiation and elongation, with the latter being important for promoting DNA repair. Our data provide insights on the molecular mechanism of ppGpp-mediated adaptation during stress, and further highlight the intricate relationships between genome stability, stress responses and transcription.

Bacteria rapidly accumulate second messenger guanosine-3′,5′-(bis)pyrophosphate (ppGpp) as a response to nutrient starvation[1]. This alarmone turns on the so-called stringent response, which shuts down transcription of ribosomal and transfer RNA and causes other metabolic changes to adjust to adverse environmental conditions[2,3]. In *Escherichia coli*, these adaptations are predominantly driven by changes in transcription initiation caused by synergetic interactions of ppGpp and transcription factor DksA with RNAP[4,5].

Independently of stringent response, ppGpp preserves genome integrity by reducing the detrimental transcription-replication collisions and by stimulating transcription-coupled DNA repair (TCR)[6–9]. Genotoxic stress surges the concentration of ppGpp, which cooperates with UvrD helicase in pulling RNAP away from the DNA damage sites, thereby exposing the lesions to the nucleotide excision repair (NER) machinery[7]. In *E. coli*, RNAP serves as a genome-wide sensor for bulky DNA lesions and as the platform to assemble the functional TCR complex comprising the NER factors UvrA, UvrB and UvrD[10,11]. TCR involves a fine balance between parallel processes involving pro-backtracking factors, such as UvrD, ppGpp and NusA[7,12], and anti-backtracking factors, such as the translocase Mfd[10,11,13,14], transcript cleavage factors GreA and GreB[12], and the leading ribosome[12,15].

How a small molecule ppGpp promotes the same large macromolecule, RNAP, to manage mechanistically different activities, transcription initiation and elongation, has remained elusive. Previous studies of the transcription initiation complex (IC) mapped three ppGpp-binding regions on RNAP: an independent Site 1, DksA-dependent Site 2, which were both ascribed to function during transcription initiation[16,17], and Site 3, observed in a cocrystal of the *Thermus thermophilus* enzyme[18], whose physiological importance has been challenged[19]. Here, we dissected the effect of ppGpp on elongating RNAP by determining high-resolution cryo-EM structures of the *E. coli* transcription elongation complex (EC) and uncoupling

[1]Department of Biochemistry and Molecular Pharmacology, New York University School of Medicine, New York, NY, USA. [2]Engelhardt Institute of Molecular Biology, Russian Academy of Science, Center for Precision Genome Editing and Genetic Technologies for Biomedicine, Moscow, Russia. [3]Howard Hughes Medical Institute, New York University School of Medicine, New York, NY, USA. [4]These authors contributed equally: Jacob W. Weaver, Sergey Proshkin, Wenqian Duan. [5]These authors jointly supervised this work: Alexander Serganov, Evgeny Nudler. ✉e-mail: alexander.serganov@nyulangone.org; evgeny.nudler@nyulangone.org

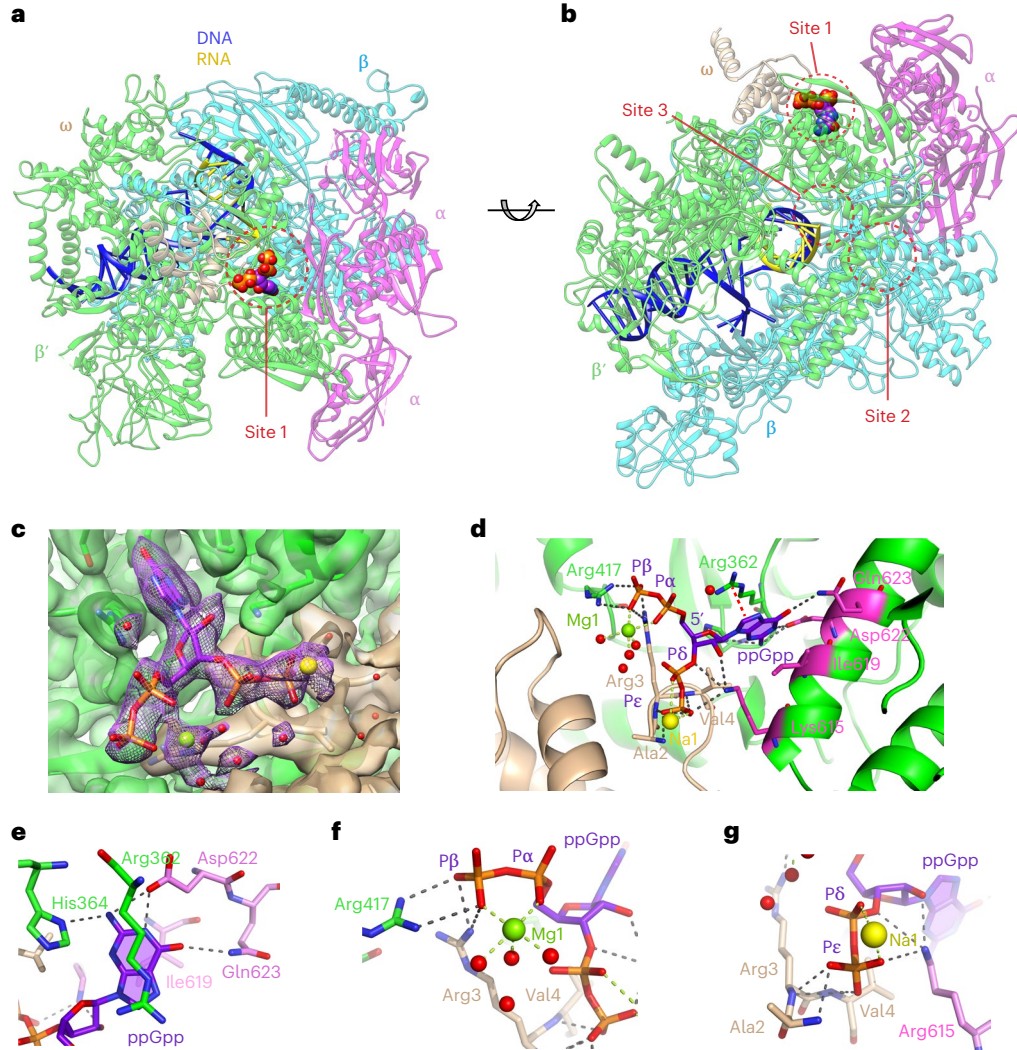

**Fig. 1 | Cryo-EM structure of the *E. coli* transcription EC bound to ppGpp.**
**a,b**, Overall structure, side (**a**) and bottom (**b**) views. ppGpp is in space-filling representation and atomic colors are C, purple; O, red; P, orange; N, blue. ppGpp-binding sites observed previously are circled. **c**, Cryo-EM map of the ppGpp-binding pocket. The map for ppGpp and adjacent $Mg^{2+}$ cation (Mg1, green sphere), $Na^+$ cation (Na1, yellow sphere) and water molecules (red spheres) is in violet mesh. **d**, ppGpp-binding pocket. Residues interacting with ppGpp are in sticks. Putative intermolecular hydrogen (distances <3.5 Å), metal coordination bonds and cation–π interactions are depicted in gray, green and red dashed lines, respectively. Amino acids altered in the Site 1 KIDQ mutant are in magenta. **e–g**, Details of ppGpp binding to RNAP depicting recognition of the nucleobase (**e**), 5′-diphosphate (**f**) and 3′-diphosphate (**g**).

functions of ppGpp in stringent response and genome integrity at the molecular level. Our data address the long-standing question of interaction of ppGpp with RNAP and provide important insights on the multifaceted role of ppGpp in bacteria.

## Results

### Cryo-EM structures reveal a ppGpp-binding site on EC

Increased level of *E. coli* RNAP backtracking upon ppGpp addition in a purified system in vitro suggested direct modulation of RNAP activity by ppGpp during the elongation phase of transcription[7]. To visualize the binding of ppGpp to elongating RNAP, we conducted cryo-EM studies on a ppGpp-bound EC, consisting of the five-subunit *E. coli* RNAP ($α_2ββ′ω$) bound to a nucleic acid scaffold mimicking a transcription bubble. The scaffold contains upstream and downstream DNA duplexes interrupted by a 10-nucleotide noncomplementary internal region where template DNA strand binds 16-nucelotide RNA.

The resulting three-dimensional cryo-EM map has a nominal resolution of 2.1 Å (Fig. 1a,b, Extended Data Fig. 1 and Table 1). The cryo-EM map for the RNAP core is well defined and has an estimated local resolution in the range 2.1–3.0 Å, with a resolution of 2.1–2.5 Å for most of the complex. A majority of RNAP and parts of the nucleic acid scaffold were fit directly into the high-resolution parts of the map. The downstream DNA clamp region consisting of the pincer tips of β and β′ subunits corresponded to the region of the cryo-EM map with the lowest local resolution of ~4 Å. The RNAP structure reveals a posttranslocated state, with the 3′ end of the RNA in the −1 position.

The extra density map between the β′ and ω subunits was assigned to ppGpp (Fig. 1c) and unambiguously confirmed by the structure of EC without ppGpp, determined in parallel on the basis of the cryo-EM map with a nominal resolution of 2.42 Å (Extended Data Fig. 2). The ppGpp-binding site on EC coincided with Site 1 previously identified on the RNAP at the transcription initiation state (Fig. 1a,b)[16,17,20].

### RNA polymerase specifically recognizes all moieties of ppGpp

The high-quality cryo-EM map (Fig. 1c and Extended Data Fig. 3) allowed for unambiguous modeling of the downstream DNA, RNA and ppGpp bound to metal cations. Octahedral coordination geometry and ~2.2-Å coordination distances (Extended Data Fig. 3c) strongly suggest

that the hydrated $Mg^{2+}$ cation binds to both phosphates of the 5′ moiety. A metal cation binding to the 3′-bisphosphate was assigned to $Na^+$ on the basis of longer coordination bonds.

ppGpp binds in the crevice formed by α, ω and β′ subunits and interacts with amino acids from β′ and ω (Fig. 1d and Extended Data Fig. 4a). The nucleobase of ppGpp nestles into a small pocket on the surface of β′ between Ile 619 and Arg 362 and makes cation–π interactions with Arg 362 (Fig. 1d,e). Three amino acids of RNAP, β′ Asp 622, Gln 623 and His 364, specifically recognize all three functionalities on the Watson–Crick edge of the nucleobase. The combination of a tight pocket, pseudo-base stacking and nucleobase-specific hydrogen bonding helps to discriminate against nonguanine nucleotides and polyphosphates.

Adding to specificity and affinity, all four phosphates of ppGpp form bonds with both ω and β′ subunits of RNAP (Fig. 1d). While the 5′-bisphosphate interacts with flexible side chains (Fig. 1f), the terminal phosphate of the 3′-bisphosphate engages main chain functionalities of ω Ala 2, Arg 3 and Val 4 (Fig. 1g). Such recognition allows the RNAP to discriminate against GTP, GDP and RNAs containing these nucleotides on the 5′ end. Interactions between ppGpp and RNAP in Site 1 are similar in the ppGpp–EC, ppGpp–IC and ppGpp–IC–DksA structures[16,17,21], albeit with some notable differences (Extended Data Fig. 4b–d and Supplementary Text). However, the biological relevance of ppGpp binding to Site 1 in ICs remains unclear.

To reveal whether the ppGpp binding affected the EC structure, we compared the structures of ppGpp-bound and -unbound ECs. The structural alignment centered on the core of RNAP indicated very similar conformations, with an overall root mean squared deviation (r.m.s.d.) for the main chain atoms of 0.23 Å. The cores were practically identical, with 0.01 degrees rotation and 0.02-Å coordinate displacement calculated using Cα atoms. Conformational differences observed in more dynamic peripheral parts of RNAP could not be reliably quantified because of the lower resolution of the map.

Inspection of the RNAP regions near the ppGpp-binding site revealed changes in the position of side chains of β′ Arg 362, Arg 417, Lys 615 and ω Arg 3 (Extended Data Figs. 3e,f, 4e) and small overall movement (2.94 degrees and 0.57 Å) of the ω subunit towards the β′ subunit, which is probably facilitated by the interactions formed between the N-terminal residues of the subunit and the sugar–phosphate moiety of ppGpp. This movement, however, does not appear to translate into conformational changes in the catalytic site, as exemplified by negligible differences in the bridge helices (0.54 degrees and 0.16 Å) and almost identical interactions with the nucleic acid scaffold (Extended Data Fig. 3a). These results suggest that ppGpp binding to Site 1 likely affects motions of RNAP during the catalytic cycle and that static three-dimensional scaffold-based structures do not capture these effects.

## Site 1 is critical for ppGpp-dependent NER

Newly observed structural details can inform on ppGpp–RNAP interactions contributing to DNA repair and stringent response. Therefore, we prepared a series of *E. coli* strains carrying mutations in Sites 1 and 2 of RNAP (Extended Data Table 1) and subjected them to ultraviolet (UV) radiation or genotoxic chemicals nitrofurazone (NFZ) and 4-nitroquinoline 1-oxide, which are known to trigger NER[22–24]. We also used phleomycin, which triggers double strand break repair that relies on RNAP backtracking[25]. Previous mutations designed to disrupt ppGpp binding[26,27] (β′ Arg362Ala Arg417Ala Lys615Ala, ωΔ2–5) were based on low-resolution structures and targeted amino acids that bind to the sugar–phosphate backbone of ppGpp (hence called here Site 1B mutant) but did not affect interactions with its nucleobase. Therefore, we have designed a more optimal mutation for eliminating ppGpp–RNAP binding by substituting Ile 619, Asp 622, Gln 623 and Lys 615 with alanines (Fig. 1d and Extended Data Fig. 5). This mutant, designated as KIDQ, prevents recognition of the nucleobase, the sugar

**Table 1 | Cryo-EM data collection, refinement and validation statistics**

| | EC (EMD-29491) (PDB 8FVR) | EC–ppGpp (EMD-29494) (PDB 8FVW) |
|---|---|---|
| **Data collection and processing** | | |
| Magnification | ×105,000 | ×105,000 |
| Voltage (kV) | 300 | 300 |
| Electron exposure (e⁻/Å²) | 59.07 | 57.54 |
| Defocus range (μm) | 0.9–1.9 | 0.9–1.9 |
| Pixel size (Å) | 0.825 | 0.825 |
| Symmetry imposed | C1 | C1 |
| Initial particle images (no.) | 786,069 | 1,001,665 |
| Final particle images (no.) | 212,142 | 643,682 |
| Map resolution (Å) | 2.42 | 2.10 |
| FSC threshold | 0.143 | 0.143 |
| Map resolution range (Å) | 2.14–39.03 | 1.82–27.19 |
| **Refinement** | | |
| Initial model used (PDB code) | 8FVW | 6ALH |
| Model resolution (Å) | 2.4 | 2.1 |
| FSC threshold | 0.143 | 0.143 |
| Map sharpening *B* factor (Å²) | −66.5 | −58.7 |
| Model composition | | |
| Non-hydrogen atoms | 26,040 | 26,108 |
| Protein residues | 3,165 | 3,165 |
| Nucleotides | 51 | 51 |
| Ligands | | |
| ppGpp | – | 1 |
| Metal cations | 3 | 5 |
| Water | 201 | 225 |
| *B* factors (Å²) | | |
| Protein | 74.89 | 56.72 |
| Nucleic acids | 79.27 | 73.96 |
| Ligands | 95.38 | 74.49 |
| Water | 44.75 | 27.55 |
| R.m.s. deviations | | |
| Bond lengths (Å) | 0.003 | 0.003 |
| Bond angles (°) | 0.643 | 0.597 |
| **Validation** | | |
| MolProbity score | 1.57 | 1.48 |
| Clashscore | 8.83 | 8.97 |
| Poor rotamers (%) | 0.07 | 0.07 |
| Ramachandran plot | | |
| Favored (%) | 97.55 | 98.06 |
| Allowed (%) | 2.45 | 1.94 |
| Disallowed (%) | 0.00 | 0.00 |

and the 3′-phosphates of ppGpp, and eliminates ppGpp binding to the EC (Extended Data Fig. 5a,b).

Viability of the parent wild-type (WT) strain moderately decreased under UV treatment and remained unchanged with NFZ treatment, while, as previously reported[7], genetic inactivation of *relA* and *spoT* (ppGpp⁰), the two ppGpp synthetases, markedly decreased survival

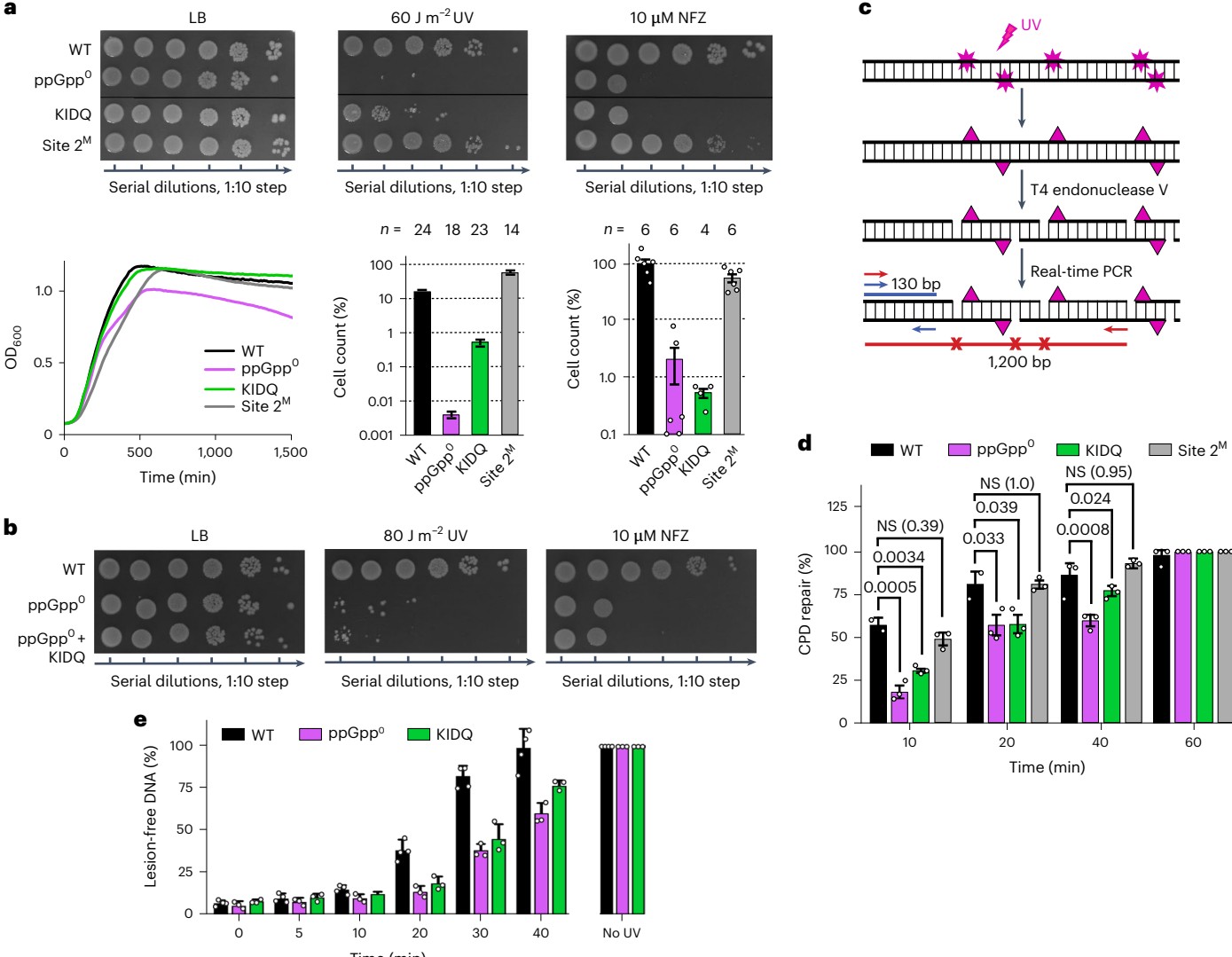

**Fig. 2 | ppGpp-binding Site 1 is necessary for efficient DNA repair. a**, Effect of genotoxic stresses on viability of WT and mutated *E. coli* cells. Bacterial cultures at mid-log phase were either treated with NFZ or irradiated by UV, serially diluted, plated on LB agar plates and individual colonies were counted. Top: representative plates. Bottom left: growth curves of bacteria in rich (LB) medium without stresses. The data points are averages of optical density (OD) values at 600 nm wavelength from eight independent growth curve measurements. A margin of error was less than 10%. Bottom middle and right: quantifications of colonies for stresses shown on top for representative experiments. Values are mean ± s.e. normalized to bacterial growth without stresses. *n*, number of biologically independent experiments. **b**, Representative plates indicating the epistatic effect of genotoxic stresses on WT and mutated *E. coli* cells. The

experiment was repeated five times with the same result. **c**, Schematic of the assay to determine repair efficiency of the UV-induced CPD in the *lacZ* locus in live bacteria. **d**, Quantification of the repair efficiency (mean ± s.e., number of independent experiments, *n* = 3) using the assay in **c**. Statistical analysis was performed using ordinary one-way analysis of variance (ANOVA) followed by Tukey's multiple comparisons test. *P* values are indicated. NS, not significant. **e**, Kinetics of the genome-wide CPD repair quantified as the percentage of lesion-free genomic DNA after UV exposure and T4 endonuclease V cleavage for each time point relative to the UV-untreated and undigested control samples (mean ± s.e., number of independent experiments, *n* = 3). The genomic DNA and cleavage products were separated on alkali–agarose gels (Extended Data Fig. 6).

(Fig. 2a). Under DNA damage conditions, the KIDQ mutant displayed a profoundly reduced survival, comparable to that of the ppGpp[0] strain (Fig. 2a and Extended Data Fig. 5c), and substantially exceeding the deleterious effects of the suboptimal Site 1[B] mutation (Extended Data Fig. 5c,d). A slightly more potent effect of ppGpp[0] than KIDQ at lower UV doses is likely due to ppGpp targeting cellular proteins other than RNAP[28].

Additionally, combining KIDQ with ppGpp[0] did not increase the deleterious effects of DNA-damaging agents on ppGpp-deficient cells (Fig. 2b), thus demonstrating the epistatic relationship between ppGpp[0] and KIDQ with respect to DNA repair. In sharp contrast, mutations known to negatively affect ppGpp binding to DksA-dependent Site 2

(ref. 29), designated as Site 2[M] mutation, did not reduce survival of bacteria under DNA-damaging conditions, suggesting the dispensability of this site for controlling transcription elongation and DNA repair (Fig. 2a and Extended Data Fig. 5c).

To test the effect of Site 1 on DNA repair directly, we determined the kinetics of the repair of the UV-induced lesions in the *lacZ* locus in live bacteria (Fig. 2c,d). Compared to WT, ppGpp[0] and KIDQ mutants demonstrate diminished cyclobutane pyrimidine dimers (CPDs) repair within the first 40 min following UV irradiation: the timeframe coinciding with the surge of intracellular ppGpp[7]. Eventually, the rate of NER recovered, and most lesions were repaired by 1 h after UV exposure. Accordingly, ppGpp[0] and KIDQ mutant cells display similarly delayed

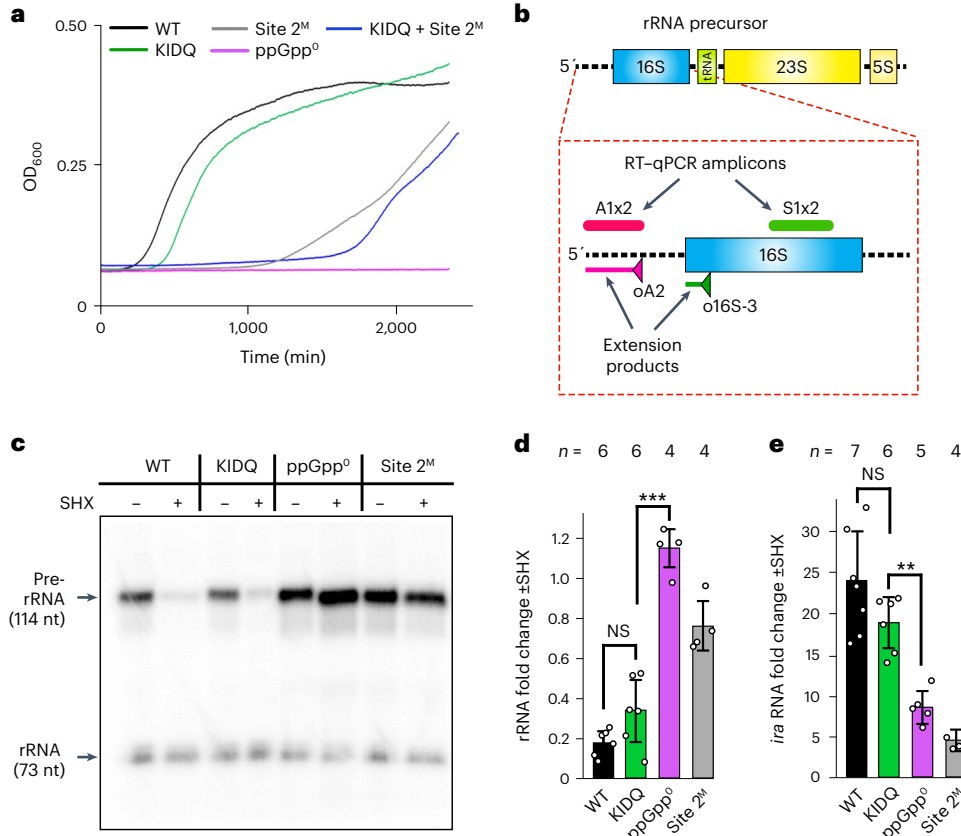

**Fig. 3 | ppGpp-binding Site 1 is dispensable for stringent response. a**, Survival of *E. coli* cells under stringent response conditions (minimal M9 medium lacking amino acids). Growth curves plotted with averaged data points (number of independent experiments, *n* = 3–8) and a margin of error less than 10%. **b**, Schematics of a ribosomal RNA precursor. Stable parts of the transcript to be matured into rRNA and tRNA are shown as rectangles. Unstable and processed 5′ leader, spacers and 3′ trailer are indicated by dotted lines. Triangles indicate the primers used in the primer extension assay in panel **c**; the extension products are shown by solid lines; A1x2 and S1x2 indicate the location of amplicons used in RT–qPCR of panel **d**. **c**, Transcription of rRNA operon under induction of stringent response by SHX. Representative gel electrophoresis of the primer extension products using primers specific to unstable 5′ leader of pre-rRNA and stable mature 16S rRNA. The experiment was independently repeated four times with the same result. nt, nucleotide. **d**, RT–qPCR of the ribosomal RNA transcription on the induction of stringent response with SHX (mean ± s.e.; *n*, number of independent experiments). NS, *P* = 0.12; ***P* < 0.0005 (*P* = 6.22 × 10⁻⁷). Statistical analysis was done by one-way ANOVA and Tukey's range test in panels **d** and **e**. **e**, RT–qPCR of the RNA transcription from *iraP* promoter upon the induction of stringent response with SHX (mean ± s.e.; *n*, number of independent experiments). NS, *P* = 0.2; ***P* < 0.005 (*P* = 0.0008).

NER across the genome within the first 40 min post UV irradiation (Fig. 2e and Extended Data Fig. 6). Thus, the Site 1 mutation that eliminates ppGpp binding in EC decreases bacterial resistance to DNA damage due to the inability of mutant cells to maintain efficient TCR.

## Site 1 is dispensable for stringent response

To test the involvement of Sites 1 and 2 in stringent response, we grew mutants on minimal medium lacking amino acids. The KIDQ mutation in Site 1 did not substantially affect growth under starvation conditions, while the ppGpp⁰ strain did not survive, and the Site 2 mutant, as well as the Site 1/Site 2 combined mutant, demonstrated severely impaired growth (Fig. 3a and Extended Data Fig. 5e).

We next measured the rate of ribosomal RNA transcription upon the induction of stringent response with serine hydroxamate (SHX). To this end, we monitored the level of the unstable 5′ leader of ribosomal 16S RNA driven by the *rrn*P1 promoter[30] (Extended Data Table 2 and Fig. 3b). A stable part of 16S RNA was used for internal normalization. In contrast to mature 16S RNA, its 5′ leader precursor is rapidly degraded, thus reflecting the rRNA promoter activity. WT and Site 1 KIDQ mutant responded to SHX by strongly decreasing rRNA transcription (a hallmark of stringent response), as detected by both the primer extension and reverse transcription quantitative PCR (RT–qPCR) assays (Fig. 3c,d). In contrast, Site 2 mutant and ppGpp⁰ cells failed to do so.

Likewise, SHX strongly induced the expression from *iraP* in WT and KIDQ mutant, but failed to do so in Site 2 and ppGpp⁰ mutants (Fig. 3e).

Together, these data show that Site 1 does not contribute to stringent response upon amino acid starvation, while Site 2 is critical for the response, as previously reported[29].

## ppGpp controls transcription elongation via Site 1

To investigate the effects of ppGpp on transcription elongation directly, we reconstituted single-round transcription runoff assays with WT and KIDQ mutant RNAPs. The WT RNAP decreased full-length (FL) transcription and prolonged pausing at discrete positions (for example, at P) in the presence of ppGpp (Fig. 4a, left panel, compare lanes 5 and 9). Such a transcriptional slowdown is expected to prompt RNAP to backtrack and facilitate TCR[7]. Indeed, the ppGpp-sensitive pauses we detected in this experiment were prone to backtracking, as confirmed by their sensitivity to the transcript cleavage factor GreB (Fig. 4b). In contrast, the KIDQ mutant RNAP was practically unresponsive to the ppGpp-induced pausing, with or without NusG and DksA (Fig. 4a, right panel, and Extended Data Fig. 7). The mutant RNAP elongated slightly faster and paused less than WT RNAP, emphasizing that ppGpp binds to a perturbation-sensitive pocket (Site 1), which is ~26 Å away from the catalytic center, to allosterically modulate transcription elongation. Despite these differences in the in vitro behavior of RNAPs, the KIDQ

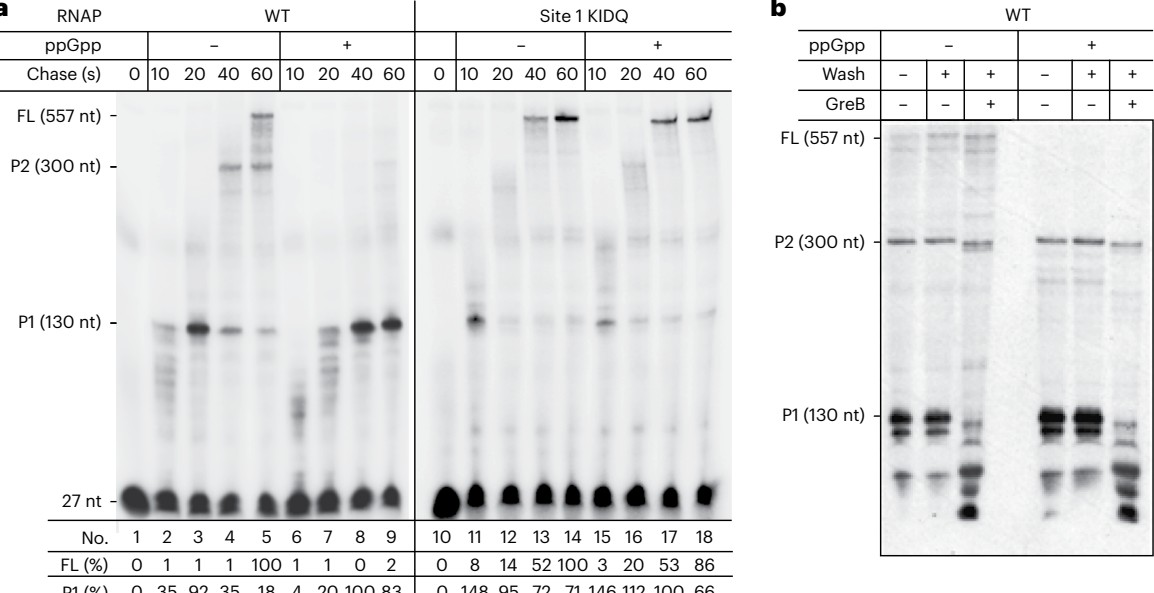

**Fig. 4 | Effect of ppGpp on transcription elongation. a**, Kinetics of single-round transcription with WT (left) and mutant (right) RNAPs in the absence and presence of ppGpp analyzed by PAGE. FL, full-length RNA; P1 and P2, major transcriptional pauses. Bottom: data for FL and P1 were normalized to the corresponding bands in lanes 5 and 14 (FL), and lanes 8 and 17 (P1). Similar values were obtained in three independent experiments with a margin of error of less than 10%. **b**, ppGpp-sensitive pauses are prone to backtracking. Paused ECs from **a** were immobilized on Ni[2+]-agarose beads, chased in the presence of ppGpp, washed and treated with GreB to reveal the extent and distance of backtracking. Similar values were obtained in two independent experiments with a margin of error of less than 10%.

mutant has no obvious growth defects, no effect on stringent response and is fully epistatic with ppGpp[0] mutation with respect to genotoxic stress, arguing that KIDQ phenotype in vivo is primarily accounted for its inability to bind ppGpp and to control RNAP elongation via Site 1.

## Discussion

The function of ppGpp in bacteria extends far beyond the classical stringent response and modulation of transcription initiation. For instance, ppGpp[0] cells exhibit high temperature sensitivity, compromised division and motility[31–34]. ppGpp deficiency has also been linked to transcription-replication conflicts, genome instability and diminished TCR[6–9,35]. None of these phenotypes has been explained in terms of promoter-specific control of RNAP. On the other hand, the ability of ppGpp to enhance RNAP pausing and backtracking[7,36] provides the means for promoter-independent regulation of transcription and its coupling to replication, translation and DNA repair[37–39].

Here we untangled the effect of ppGpp on RNAP initiation and elongation. We show that the latter is controlled via Site 1: a ppGpp-binding pocket of hitherto unknown function. Thus, the two distinct ppGpp-binding sites on RNAP, Site 1 and Site 2, serve to control transcription elongation and initiation, respectively (Fig. 5). We further show that it is Site 1 that accounts for the ability of ppGpp to stimulate TCR. Our biochemical data show that ppGpp promotes backtracking of RNAP through Site 1, that is, it supports RNAP functioning as a chromosomal damage scanner[7,10,11,40]. Thus, unlike Site 2, which requires DksA for ppGpp binding[4,5], Site 1 represents an attractive new target for rational design of small-molecule modulators of transcription with potential antimicrobial and industrial microbiology applications.

A commonality to regulation of transcription elongation often involves conformational rearrangement of RNAP. This has been seen with protein factors such as NusA[41,42], RfaH[43] and λQ[44]. Here we demonstrate an endogenous small molecule that is capable of affecting RNAP in a similar manner to bona fide elongation factors. Interestingly, ppGpp binding does not induce substantial conformational

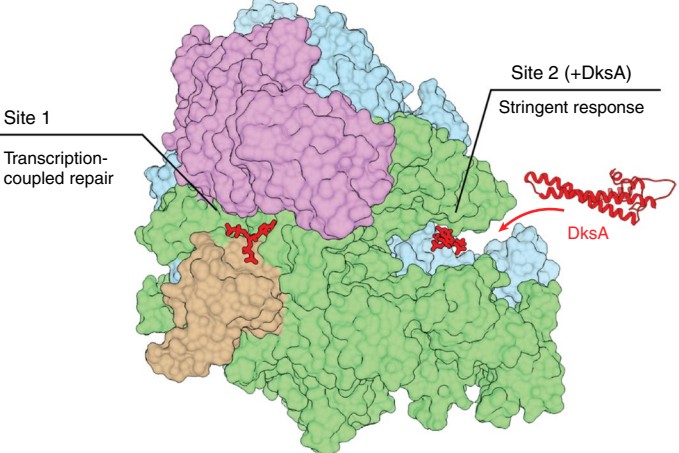

**Fig. 5 | Distinct effects of ppGpp binding to RNAP.** ppGpp binds to two separate sites on RNAP affecting different processes depending on the site. Transcription-coupled repair is enhanced by ppGpp binding to Site 1 during transcription elongation. Stringent response is triggered by ppGpp and DksA binding to Site 2 during transcription initiation.

changes in RNAP in the cryo-EM structure, offering a possibility that the small molecule makes RNAP more prone to backtracking via affecting motions of RNAP during the elongation cycle. Since mismatches in the transcription bubble of the nucleic acid scaffold would not allow backtracking, the EC–ppGpp structure does not capture the backtracking conformation.

By responding to ppGpp, RNAP behaves as a molecular sensor that monitors ppGpp levels as a readout for a metabolic state[45]. ppGpp rises under stresses, such as DNA damage, amino acid starvation or antibiotics[7,46]. The same stressors often result in suppressor mutations in

ppGpp[0] cells localized to the vicinity of Site 1 and other RNAP domains involved in modulation of the elongation rate[47–50], suggesting that ppGpp-mediated control of transcription elongation is a multipurpose regulatory hub. Indeed, our recent results indicate that ppGpp exerts its protective effect against antibiotic stress and cell division defects by decelerating transcription elongation in vivo[51].

The regulation by ppGpp via Site 1 described here for *E. coli* must be conserved among proteobacteria[1]. Other bacteria may also unitize ppGpp for potentiating backtracking and TCR, as RNAP from evolutionarily distant species, such as Mycobacterium species, responds to the alarmone during elongation[51]. Indeed, the overall architecture of cellular RNAPs and the phenomena of backtracking and TCR are preserved in evolution[40]. The discovery of RelA/SpoT homologs in eukaryotes implies the presence of ppGpp in higher organisms as well[52–54]. The role of ppGpp in sleep control in *Drosophila* has recently been reported[55]. Given the multitude of nuclear processes coupled to transcription elongation[56], this type of allosteric control of moving RNAP could be a regulatory nexus for messenger RNA biogenesis, chromatin modification and genomic integrity.

## Online content

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

## Methods

### Preparation of elongation complexes

The elongation complex contained a three-strand oligonucleotide scaffold[57] composed of two 53-nucleotide DNA strands and 16-nucleotide RNA obtained by annealing of oligonucleotides ntDNA, tDNA and oRNA (Extended Data Table 2), synthesized by Integrated DNA Technologies. DNAs have extended regions of complementarity to form a double-stranded DNA template interrupted by a 10-nucleotide stretch of nucleotides incapable of base pairing to form an artificial transcription bubble (Extended Data Fig. 3a). EC was formed as in ref. [57]. Briefly, tDNA and oRNA were mixed in a 1:1 molar ratio, heated at 95 °C for 2 min, 75 °C for 2 min, 45 °C for 5 min and cooled down to 25 °C slowly. The annealed tDNA–oRNA hybrid was aliquoted and stored at −20 °C. The hybrid was added in a 1.2-fold molar excess to RNA polymerase and incubated at room temperature for 15 min. A fivefold molar excess of ntDNA was added to the complex and the mixture was incubated for an additional 10 min. The elongation complex was purified by size exclusion chromatography (Superdex 200 Increase 10/300 GL; GE Healthcare) in EC buffer (20 mM HEPES, pH 8.0, 150 mM NaCl, 10 mM $MgCl_2$, 10 μM $ZnCl_2$, 5 mM DTT) and concentrated before use.

### Cryo-EM

The EC sample for cryo-EM experiments was diluted to 6.5 mg ml$^{-1}$ and supplemented with 8 mM CHAPSO[58]. For EC–ppGpp samples, EC was mixed with 0.1 mM ppGpp (TriLink) for 20 min at room temperature, followed by addition of 8 mM CHAPSO. Quantifoil R 300 mesh 1.2/1.3 holey gold grids were glow discharged in the PELCO easiGlow glow discharge cleaning system for 25 s. A 3.5-μl portion of EC or EC–ppGpp complex was applied to grids, blotted with a blot force of 2 and a blot time of 3 s, and plunge frozen into liquid ethane with the chamber at 100% humidity and 10 °C temperature using a Vitrobot Mark IV.

The cryo-EM data were collected using a 300-kV Titan Krios (ThermoFisher Scientific) equipped with a K3 Summit direct electron detector at a magnification of ×105,000, corresponding to a pixel size of 0.825 Å. Datasets were collected with Leginon (v.3.5)[59] at an accumulated dose of 57.54 e$^-$/Å$^2$ (EC–ppGpp) and 59.07 e$^-$/Å$^2$ (EC). Intermediate frames were recorded every 0.04 s for a total of 50 frames per micrograph. A total of 4,921 images (EC–ppGpp) and 3,298 images (EC) were collected at nominal defocus range of 0.9–1.9 μm and ice thickness range of 40–105 nm (ref. [60]).

The datasets were subjected to beam-induced movement correction using MotionCor2 (ref. [61]) on the Appion pipeline[62] and imported to cryoSPARC[63]. Contrast transfer function (CTF) was estimated using Patch CTF in cryoSPARC (v.3.3.2). In total, 786,069 (EC) and 1,001,665 (EC–ppGpp) particles were picked in template-free mode using blob picker and extracted with a 100-pixel box after downscaling the pixel size to 3.3 Å. Several rounds of two-dimensional (2D) classifications were performed to remove junk particles. These particles were rescaled to the original pixel size and filtered by another cycle of 2D classification. In total, 212,142 (EC) and 643,682 (EC–ppGpp) particles were used in cryoSPARC for ab initio reconstruction of initial three-dimensional (3D) models and further 'homogeneous' and 'nonuniform' refinements[64]. CTF parameters were refined on a per particle basis using cryoSPARC local CTF refinement. Particles were once more subjected to 'homogenous' refinement, resulting in the maps with a global nominal resolution of 2.10 Å (EC–ppGpp) and 2.42 Å (EC) at a Fourier shell correlation (FSC) of 0.143. Final maps were autosharpened with the $B$ factors listed in Table 1, using cryoSPARC. Local resolution was estimated using cryoSPARC.

### Model building, refinement and analysis

A previously determined EC structure (PDB 6ALH) was rigid-body fit into the cryo-EM maps using UCSF Chimera[65]. For the EC–ppGpp structure, the protein model was docked into the map using Phenix (v.1.17.1)[66,67] while the nucleic acid scaffold was manually built using

Coot (v.0.8.9.2)[68]. The resulting structural model was refined using the sharpened map and real-space refinement in Phenix[66] and Coot. The EC structure was built using the structure of EC–ppGpp. The EC–ppGpp structure was docked into the sharpened EC map and manually adjusted in Coot for the best fit, followed by rounds of real-space refinement in Coot and Phenix. ppGpp was added into the unaccounted map on the basis of the difference between the EC–ppGpp and EC maps.

Water molecules were added to the EC–ppGpp reconstruction using the unsharpened map and the 'phenix.douse' routine with default parameters and a 2.31–3.20-Å bond distance. All water molecules were manually checked in Coot. Molecules placed near low-resolution regions were removed and extra molecules were placed in the high-resolution regions on the basis of the round shape of the map peaks and hydrogen bonding geometry and distances. $Mg^{2+}$ and $Na^+$ cations were added on the basis of the coordination geometry and coordination distances. $Zn^{2+}$ cations were added according to the published structures.

Water molecules were added to the EC reconstruction in two steps. First, the molecules were added de novo using the unsharpened map and 'phenix.douse' and verified manually in Coot. The molecules near low-resolution regions were removed. Second, additional water molecules were placed on the basis of the projection of the water molecules from the EC–ppGpp structure onto the sharpened EC map.

The structures were additionally real-space refined. Model quality and model-versus-data fit statistics were evaluated using MolProbity[69,70]. Structural figures were generated by UCSF Chimera v.1.15 (ref. [65]) and PyMOL (The PyMOL Molecular Graphics System, v.2.5.4, Schrödinger, LLC).

Structure superpositions and r.m.s.d. calculations were done for Cα atoms using the Superpose program[71] from the CCP4 package[72]. The core module was defined as in ref. [73]: α 1–234; β 10–26, 514–828, 1071–1235; β' 504–771. For determining the angle and displacement values between structures, the 'align' routine of PyMOL was used to superpose Cα atoms of the core modules of ECs. Then, the angle and displacement values were calculated for structural elements using the 'angle_between_domain' routine contained within 'orientation. py' of PyMOL.

### Bacterial strains and plasmids

*E. coli* MG1655 strains and plasmids used in this study are listed in Extended Data Table 1. *E. coli* strains with substitutions in ppGpp-binding sites were constructed by using the lambda Red recombineering method together with CRISPR–Cas9 counterselection[74].

Oligonucleotides (~80-mers) used for recombination were designed to target the lagging strand of replicating DNA with upstream and downstream homology to the area of mutagenesis. Three phosphorothioated bases were incorporated at the 5' end of the oligonucleotides to reduce their degradation rate in vivo. pKDsg derivatives (pSg-615, pSg-619 and pSg-680 plasmids) used for counterselection were constructed by circular polymerase extension cloning using primers with overlapping 20-bp protospacer sequences that corresponded to the fragment adjacent to appropriate protospacer adjacent motif sites (5'-NGG) in the parental sequences.

*E. coli* strain with β' K615A substitution was constructed at the first step. Wild-type MG1655 strain was first transformed with the pCas9cr4 plasmid and subsequently transformed with the single guide RNA (sgRNA)-encoding plasmid (pSg-615). Cells that possessed both plasmids were grown in SOB medium with spectinomycin (Sp, 50 mg l$^{-1}$) and chloramphenicol (Cm, 30 mg l$^{-1}$) at 30 °C. At OD$_{600}$ ≈ 0.5, lambda Red was induced with 1.2% (w/v) L-arabinose, and cells were grown for another 20 min. Then, a mutagenic oligonucleotide for recombineering was electroporated into the cells. After 2 h of recovery, the cells were plated on LB medium supplemented with Sp, Cm and anhydrotetracycline (aTc, 100 ng l$^{-1}$) and incubated overnight at 30 °C to select for survivors of the CRISPR–Cas9 selection. Colonies were screened with

specific primers to identify mutated regions, and the corresponding chromosome region was verified by sequencing. To eliminate the pSg-615 plasmid, cells were incubated in LB for 12 h at 37 °C and streaked on LB plates. Individual colonies were selected and assessed for the loss of Sp resistance.

The β′ I619A, D622A and Q623A substitutions were generated in one step by the same procedure using pSg-619 plasmid for selection and the K615A strain as recipient. The resulting strain contained four mutations, β′ K615A, I619A, D622A and Q623A, designated as the KIDQ mutant.

Mutations encoding β′ N680A and K681A (Site 2[M] mutant) were introduced to the MG1655 pCas9cr strain as described above for the Site 1 mutants, with the use of a corresponding mutagenic oligonucleotide and the pSg-680 plasmid encoding appropriate sgRNA. All strains were sequenced to confirm desired mutations.

To reproduce the Site 1[B] mutant, four mutagenesis steps were made as described above to prepare β′ K615A, a deletion of the ω N-terminal residues 2–5, β′ R417A and β′ R362A in the MG1655 strain. On each step, an appropriate sgRNA-encoding plasmid and a mutagenic synthetic oligonucleotide were used. Strains constructed as intermediates were cured of a preceding pSg plasmid and immediately transformed with a new relevant plasmid to introduce an additional mutation.

The KIDQ mutations were introduced into the strain *relA256 spoT212* and Site 2[M] mutant in a one-step recombination process using double-stranded DNA that encoded all four (β′ K615A, I619A, D622A and Q623A) substitutions.

The pCas9cr plasmid was cured by the pKDsg-15a plasmid that targeted the p15a origin of replication of pCas9cr. Upon transformation of pKDsg-15a into cells that contained pCas9cr, the cells were recovered in SOC medium for 2 h at 30 °C, followed by addition of aTc (100 ng l[−1]) and additional incubation for an additional 2 h, before plating on LB/agar plates supplemented with Sp and aTc. The pKDsg-15a plasmid was cured by growth at 37 °C.

To create the pSP-KIDQ plasmid, a fragment of the *rpoC* gene with mutations encoding β′ K615A, I619A, D622 and Q623 substitutions was amplified from genomic DNA of the SP1231 strain and inserted into the SbfI and KflI sites of the pVS10 plasmid to replace the corresponding region of the wild-type gene.

*Streptomyces morookaensis* strain CF16775 was obtained from M. Cashel[75].

### Generation of growth curves
Mutant and WT *E. coli* strains were grown in LB, pelleted, washed twice in M9 minimal medium and resuspended to $OD_{600} \approx 0.01$ in LB or M9 medium lacking amino acids. A 150-μl portion of each diluted culture was pipetted into the honeycomb wells and grown at 37 °C with vigorous shaking on the platform of the Bioscreen C MBR automated growth analysis system. $OD_{600}$ values were recorded automatically every 10 min and plotted. The curves were generated on the basis of averaged values ($n = 8$) with a margin of error of less than 10%.

### Analysis of rRNA leader and *iraP* transcripts level
rRNA promoter activity was analyzed as in ref. 29 by monitoring the synthesis of the unstable leader region generated by the *rrsB* P1 promoters. *E. coli* strains were grown at 37 °C in LB to stationary phase, diluted to $OD_{600} \approx 0.04$ and allowed to grow to an early exponential phase ($OD_{600} = 0.2$–0.3). Each test culture was split into 10-ml aliquots and an aliquot received serine hydroxamate (Sigma) to 1 mg ml[−1] to induce amino acid starvation. After 15 min of incubation at 37 °C, cultures were mixed with an ice-cold ethanol/phenol stop solution to inactivate cellular RNases[76], pelleted, washed by cold saline and stored at −80 °C. Total RNA was extracted by the hot phenol method[76], and integrity of 16S and 23S rRNAs was verified by Qubit RNA IQ assay kit and checked by agarose gel electrophoresis. RNA quantitation was made by Qubit RNA BR assay kit. RNA was treated with ezDNase (Invitrogen) for the

fast removal of contaminating genomic DNA and 0.25 μg of total RNA of each sample was used as a template for complementary DNA synthesis using SuperScript IV reverse transcriptase (Invitrogen). RT–qPCR was performed with qPCRmix-HS SYBR (Evrogen) on a QuantStudio 5 Real-Time qPCR machine (Applied Biosystems). rRNA leader levels originated from the P1 promoter of the *rrn* operons were determined using the same primers as in ref. 29, to generate a 112-bp product (Extended Data Table 2). To determine the *iraP* promoter activity, a pair of primers that correspond to the beginning of the *iraP* transcript were used. A 98-bp product corresponding to the 3′ end of 16S rRNA was used for normalization of the RNA amount in the samples. The fold change in expression of the rRNA leader or *iraP* RNA after amino acid starvation relative to the internal control (16S rRNA) was calculated with the Pfaffl method[77]. Final values are averages from at least four experiments ($n = 4$–7).

### Primer extension assay for stringent control
Primer extension was performed with [32P]-labeled primers oA2 and o16S-3. The primer oA2 is complementary to the 5′ leader RNA, positions +90 to +114 relative to the start of transcription. The primer o16S-3 hybridizes to a stable part of 16S RNA, positions +52 to +73, downstream from the mature 5′ end, and was used as an internal reference. Total RNA (10 μg) was mixed in water with 2 μl radiolabeled primers (3 pmol of oA2 and 0.04 pmol of o16S-3) in a volume of 12 μl. Note that the amount of o16S-3 primer was 75-fold less than the amount of oA2 to prevent overexposure of extension products. Samples were heated at 80 °C for 3 min and chilled on ice. Annealed RNA was combined with RT reaction mix containing 1× SuperScript IV buffer, 5 mM DTT, 0.5 mM dNTP mix, 20 U SUPERase-In (Ambion) and 200 U SuperScript IV reverse transcriptase (Invitrogen) and reaction proceeded at 53 °C for 30 min. The reaction was stopped by adding 10 μl stop solution (95% (v/v) formamide, 20 mM EDTA, 0.03% bromophenol blue and 0.03% xylene cyanol) and heated at 90 °C for 3 min. Primer extension products were resolved on 8% denaturing polyacrylamide gel and visualized by phosphorimaging. The experiment was repeated four times with similar results.

### Preparation of radioactive ppGpp
Radiolabeled ppGpp was synthesized using pyrophosphotransferase from the extract of *Streptomyces morookaensis* strain CF16775 (ref. 75). Bacteria were grown at 28 °C in 5 ml LB with shaking at 180 r.p.m. for 18 h. A 1-ml starter culture was used to inoculate 500 ml LB. Bacteria were grown under the same conditions in the presence of 30 g of 3 mm glass beads to disperse the culture. The culture was passed through a Buchner funnel to remove glass beads. Cells were pelleted at 20,000*g* for 20 min. Then, 25 ml 1 M Tris–Cl, pH 8.0, was added to the supernatant with stirring at 4 °C. Next, the pyrophosphotransferase fraction was precipitated with 200 g of ammonium sulfate added gradually with stirring at 4 °C for 30 min. The mixture was spun at 25,000*g* and 4 °C for 30 min. The supernatant was decanted and the pellet was resuspended in 11 ml of 10 mM Tris–Cl, pH 8.0, 0.1 mM EDTA and 10% glycerol. The resuspended extract was dialyzed against 1 liter of the same buffer at 4 °C overnight and was stored at 4 °C.

The synthesis of the [32P]-labeled ppGpp was carried out in 50 μl of 50 mM Tris–HCl, pH 8.0, 0.5 mM EDTA, 20 mM $MgCl_2$, 9 mM cold ATP, 0.5 mCi [γ-32P]ATP (10 mCi mmole[−1]) (donors), 9 mM GDP and 12.5% (v:v) CF16775 extract. The reaction was incubated at 37 °C for 1 h and stopped by adding 1 volume of phenol:chloroform:isoamyl alcohol (25:24:1) mixture. The reaction was vortexed and spun at 20,000*g* for 5 min. The aqueous phase was removed and LiCl was added to 2 M. Nucleotides were precipitated by adding 5 volumes of 96% ethanol, incubating overnight at −20 °C and centrifuging at 20,000*g* for 45 min at 4 °C. The pellet was dried and resuspended in 500 μl water.

The resuspended ppGpp was passed over a 1 ml gravity flow column packed with QAE Sephadex A-25 resin equilibrated in 20 mM Tris–Cl, pH 8.0, 0.5 mM EDTA, 0.1 M LiCl. The column was washed

with 5 volumes of the same buffer followed by 5 volumes of the buffer containing 20 mM Tris–Cl, pH 8.0, and 0.3 M LiCl. ppGpp was eluted from the column with 5 volumes of the buffer containing 20 mM Tris–Cl, pH 8.0, and 0.5 M LiCl. The final purified fractions were pooled, adjusted to 3 M LiCl and precipitated with 5 volumes of 96% ethanol at −20 °C. The pellet was dried and resuspended in 50 µl of the buffer containing 20 mM Tris–Cl, pH 8.0, 0.05 mM EDTA. ppGpp purity was verified by thin-layer chromatography on PEI-cellulose plates. ppGpp concentrations were estimated by absorbance at 254 nm ($\varepsilon = 13.7$ mM$^{-1}$ cm$^{-1}$).

### Determining binding affinity of ppGpp to EC
Binding affinity between ECs and ppGpp was determined by DRaCALA, essentially as described in ref. [78]. Binding reactions were carried out in 10-µl mixtures at room temperature for 10 min using 0.1 µM radioactive ppGpp and serially diluted ECs in the 0.1–12.5 µM range. Note that concentrations >12.5 µM cause precipitation of EC. The 5-µl aliquots of the reactions were spotted on nitrocellulose membrane and allowed to dry. Dried membranes were exposed to a phosphor screen and imaged on a Typhoon Imager (Cytiva). Reactions were quantified using ImageJ[79] according to ref. [78] and fit to a one-site specific binding equation using GraphPad Prism v.9.0.0 for Mac and v.8.4.3 for Windows (GraphPad Software).

### Measuring sensitivity to DNA damage
*E. coli* strains were cultured in LB medium overnight. Cultures were diluted 100-fold and grown to OD$_{600}$ ≈ 0.4, after which 10-fold serial dilutions were spotted onto regular LB agar plates (for the representative plates shown in Fig. 2a) or spread over large LB agar plates (for numerical quantitation) containing the indicated concentrations of NFZ, followed by incubation overnight (24 h) at permissive temperatures. Colony-forming units were counted at each concentration of the genotoxic agent. NFZ was prepared as 10 mM stock solutions in *N,N*-dimethylformamide and diluted appropriately for each experiment. For UV survival assays, overnight bacterial cultures were diluted 100-fold and grown to OD$_{600}$ ≈ 0.4, after which appropriate dilutions were spread or spotted (for qualitative spotting assay) onto LB agar plates. Samples were then irradiated with a UV lamp (254 nm) and incubated overnight at appropriate temperatures. All assays were conducted with several bacterial clones. The Δ*relAspoT* strain was routinely assayed for compensatory suppressor mutations by streaking on minimal media plates. The lack of growth on minimal media plates indicated a lack of suppressor mutations.

To test growth under stringent response conditions, overnight bacterial cultures were diluted 10-fold and grown to OD$_{600}$ ≈ 0.4 in LB. Cells were collected by mild centrifugation, washed in the minimal M9 medium lacking amino acids and plated onto agar plates with the same minimal medium in tenfold serial dilutions.

### DNA repair on the isolated locus
DNA repair measurements were conducted according to the methods described in ref. [80]. Overnight bacterial cultures were diluted 50-fold in LB and grown at 37 °C to OD$_{600}$ ≈ 0.3. Cells were collected by centrifugation and resuspended in an equal volume of M9 salts. An aliquot of cells was spread on a Petri dish and irradiated with UV light (254 nm) at a dose of 40 J m$^{-2}$. The dose rate was maintained at 1 J m$^{-2}$ s$^{-1}$ (ref. [81]). The irradiated cells were diluted twofold in 2× LB medium and incubated at 37 °C for varying times. Cells were collected by centrifugation and genomic DNA was isolated using Purelink genomic DNA isolation kit (ThermoFisher Scientific). All manipulations, from UV irradiation to DNA isolation, were performed in the dark to prevent reversal of CPDs by photoreactivation.

The CDP repair efficiency was estimated at the chromosomal *lacZ* locus by a semi-long-range qPCR. To convert CPD into the DNA breaks, 2 µg of genomic DNA was treated with 4 units of T4

endonuclease V for 1 h. DNA was purified with the PCR purification kit (Qiagen) and subjected to real-time PCR analysis in the 96-well plate format in a Quantstudio 7 instrument (Applied Biosystems). The reaction mixture contained 1× Amplitaq master mix, 0.05× LightCycler 480 Resolight dye (Roche), 500 nM of each primer and 1 ng of template DNA in a total volume of 20 µl per well. The first set of primers yielded a short 131-bp amplicon (sLaZ1 and sLaZ2 primers) representative of the total amount of template (an internal normalization control). The second set of primers yielded a longer 1,147-bp amplicon (lLacZ1 and lLacZ2 primers). Since the DNA breaks abrogate PCR amplification, the amount of the long PCR product correlates with the amount of undamaged or repaired DNA. The number of DNA lesions was calculated using a modified version of the 2$^{\Delta\Delta Ct}$ method[82]. Under our UV irradiation conditions, the lesion frequency was 8–10 per 10 kilobase (kb) of genomic DNA. We assumed the number of CPDs at time 0 min to be the total number of CPDs that need to be repaired. Percentage of CPDs repaired at a given time point was calculated as a fraction of CPDs at time 0 min that remained at that time point. Each experiment was repeated three times independently.

### Genome-wide DNA repair
Estimation of the UV-induced DNA damage and repair was adapted from the procedure described in refs. [83],[84] with some modifications. To prepare cells for UV irradiation, single colonies of the *E. coli* strains were inoculated into LB medium and grown overnight at 37 °C. The overnight cultures were diluted 1:100 and the growth continued to OD$_{600}$ ≈ 0.3. The cells were centrifuged and resuspended in M9 medium (1× M9 salts, 0.4% glucose, 0.2% casamino acids, 2 mM MgSO$_4$, 0.1 mM CaCl$_2$ and 1 mM thiamine hydrochloride). The UV exposure (254 nm) was done at 100 J m$^{-2}$. The 2× LB medium was added after the UV irradiation and cells were recovered at 37 °C in the dark for various time intervals. The 1-ml aliquots of the culture were collected and mixed with prechilled 2× NET buffer (1× NET buffer: 100 mM NaCl, 10 mM Tris, pH 8.0, 20 mM EDTA, pH 8.0) along with the nonirradiated control sample stored on ice. The cells were centrifuged and used for the isolation of genomic DNA. The genomic DNA was isolated using the Master Pure Complete DNA and RNA purification kit (Lucigen) following the manufacturer's protocol. The isolated genomic DNA was dissolved in 35 µl of TE buffer (10 mM Tris–HCl, pH 7.5, 1 mM EDTA). A 15-µl aliquot of each DNA sample was treated with 3 U of T4 endonuclease V for 45 min at 37 °C. The undigested and T4-digested genomic DNA samples were then electrophoresed on 0.5% alkaline agarose gels in 30 mM NaOH, 1 mM EDTA at 25 V and room temperature for 18–20 h. The gels were stained and visualized with SYBR Gold. The intensity of each high-molecular band was determined using ImageJ software (v.1.52a) and the fraction of lesion-free DNA was quantified as a ratio of the T4-digested and undigested samples. To account for any nicks in the DNA that occurred before UV irradiation, the value at each time point was normalized to the value from the nonexposed control sample (−5 min point).

### Elongation rate measurements
Transcription elongation assay was conducted using a DNA template consisting of the T7A1 promoter and short downstream sequence fused to a portion of the *trpA* gene containing *trpT′* Rho-dependent terminator. To initiate the reaction, 7.5 pmol His-tagged core WT or mutant RNAP in 20 µL TB100 buffer (40 mM Tris–HCl, pH 8.0; 10 mM MgCl$_2$; 100 mM NaCl) was mixed with 0.2 µl of sigma 70 (28 pmol) subunit and 7.5 pmol DNA template. The mixture was incubated at 37 °C for 5 min. An RNA primer with the AUC sequence was added to the mixture to a final concentration 10 µM. GTP and ATP were added up to 25 µM and incubation continued at 37 °C for 5 min. Reactions were mixed with 2 µL α-[$^{32}$P]CTP (3,000 Ci mmol$^{-1}$; Perkin Elmer) and incubated at 22 °C for 5 min, followed by addition of 5 µM CTP and incubation for two additional minutes. Mixtures with total volume 30 µl were diluted

with 60 µl TB100, mixed with 5 µl DksA (up to 1 µM) and 5 µl NusG (up to 1 µM), and split into two 40-µl samples. One 10-µl aliquot from each sample was quenched by 10 µl of stoppage buffer (SB) (1× TBE, 20 mM EDTA; 8 M urea, 0.025% xylene cyanol and 0.025% bromophenol blue). Remaining samples were mixed with 4 µl ppGpp (up to 1 mM concentration) or water and incubated at 22 °C for 5 min. Reactions were chased by 0.1 mM NTPs for 10, 20, 40 or 60 s at 22 °C before taking 10-µl aliquots and quenching them with 10 µl SB. Quenched samples were heated at 100 °C for 5 min and loaded onto 6% (20 cm × 20 cm) 19:1 polyacrylamide gel supplemented with 7 M urea and 1× TBE. The gel was run at 50 W for 20 min. The gel was transferred onto Whatman paper and dried at 80 °C for 2 h. The gel was exposed to a phosphor screen for 60 min and visualized by a Typhoon Phosphorimager (GE Healthcare). Position of the paused sites was determined by sequencing reactions.

## Backtracking recovery assay

To initiate the reaction, 7.5 pmol core WT RNAP in 20 µl TB100 was mixed with 0.2 µl of sigma 70 (28 pmol) subunit and 7.5 pmol biotinylated DNA template. The mixture was incubated at 37 °C for 5 min. A 10 µM RNA primer with the AUC sequence was added to the mixture together with 25 µM GTP and ATP and incubation continued at 37 °C for 5 min. The resulting EC11 complex was immobilized on Neutravidin beads (~10 µl; Piers) in the presence of 1.5 mg ml$^{-1}$ heparin for 5 min at room temperature. Reactions were mixed with 2 µl α-[$^{32}$P]CTP and incubated at 22 °C for 5 min. The resulting EC20 was washed twice with 1 ml of TB1000 (TB with 1 M NaCl) and twice with 1 ml of TB100. The volume in the sample was adjusted to 30 µl by TB100 premixed with DksA (up to 1 µM), NusG (up to 1 µM) and the sample was incubated with 1 mM ppGpp for 5 min at 22 °C. EC20 was chased for 30 s at room temperature before 10-µl aliquots were withdrawn and quenched by 10 µl SB. The rest of the resin was immediately washed four times with 1 ml of TB100. The sample volume was adjusted to 20 µl by TB100 and the sample was split in two 10-µl reactions. One reaction was quenched by 10 µl of SB and the second reaction was incubated for 5 min with 1 µM GreB before quenching. The quenched samples were heated at 100 °C for 5 min and loaded onto 6% (20 cm × 20 cm) 19:1 polyacrylamide gel supplemented with 7 M urea and 1× TBE. The gel was run at 50 W for 20 min. The gel was transferred onto Whatman paper and dried at 80 °C for 2 h. The gel was exposed to an X-ray film and scanned. Experiments were repeated as described in the figure legends.

## Statistics and reproducibility

No statistical method was used to predetermine sample size. No data were excluded from the analyses. The experiments were not randomized. The investigators were not blinded to allocation during experiments and outcome assessment.

## Reporting summary

Further information on research design is available in the Nature Portfolio Reporting Summary linked to this article.

## Data availability

The atomic coordinates and cryo-EM maps from this study are deposited in the Protein Data Bank (PDB) and Electron Microscopy Data Bank (EMDB) under PDB codes 8FVR (EC) and 8FVW (EC + ppGpp), and EMDB entries EMD-29491 (EC) and EMD-29494 (EC + ppGpp), respectively. All other data are available in the manuscript or supplementary materials. Requests for strains or plasmids will be fulfilled by the lead contact author (E.N.) upon request.

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

## Acknowledgements

This work was supported by the Russian Science Foundation grant 17-74-30030 (A.M. and S.P.), the NIH grants T32 GM088118-09 (J.W.W.), 2R01GM112940 (A.S.) and R01GM126891 (E.N.), Blavatnik Family Foundation and by the Howard Hughes Medical Institute (E.N.). We thank personnel of NYU Langone Health's Cryo-Electron Microscopy Laboratory (RRID: SCR_019202), Laboratory for BioMolecular Structure (Brookhaven National Laboratory) and New York Structural Biology Center for help in data collection. The Laboratory for BioMolecular Structure (LBMS) is supported by the DOE Office of Biological and Environmental Research (KP1607011). The National Center for CryoEM Access and Training (NCCAT) and the Simons Electron Microscopy Center located at the New York Structural Biology Center are supported by the NIH Common Fund Transformative High Resolution Cryo-Electron Microscopy program (U24 GM129539), and by grants from the Simons Foundation (SF349247) and NY State Assembly.

## Author contributions

W.D. and J.W.W. determined cryo-EM structures. J.W.W. performed binding studies. S.P. and E.A. conducted genetic experiments, sensitivity tests to genotoxic stress, stringent control assay and qPCR. V.E. performed in vitro elongation experiments. M.G. and B.K.B. determined DNA repair efficiency. A.M. assisted with genetic data analysis. W.D., J.W.W and A.S. analyzed structural data. E.N. conceptualized the study, designed and supervised biochemical experiments. A.S. supervised the structural studies. J.W.W., W.D., A.S. and E.N. wrote the manuscript. All authors discussed the results.

## Competing interests

The authors declare no competing interests.

## Additional information

**Extended data** is available for this paper at https://doi.org/10.1038/s41594-023-00948-2.

**Correspondence and requests for materials** should be addressed to Alexander Serganov or Evgeny Nudler.

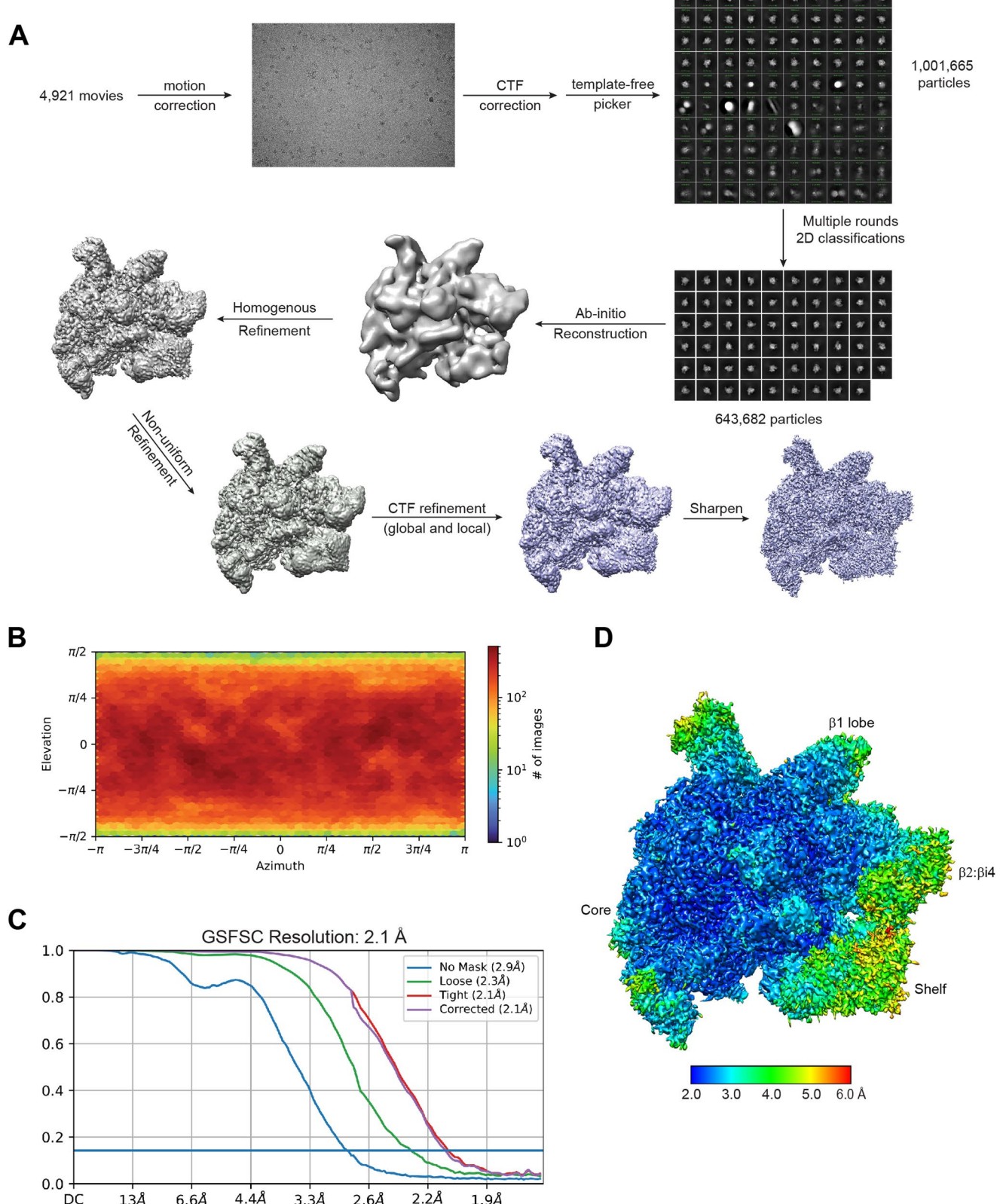

**Extended Data Fig. 1 | Cryo-EM workflow and resolution analysis for ppGpp-bound EC. a**, Processing and classification flow chart. **b**, Angular distribution of particle orientation. **c**, Fourier shell correlation (FSC) plot for half-maps with 0.143 FSC criteria indicated. The nominal resolution is determined to be 2.1 Å.

**d**, A cryo-EM map colored according to local resolution. The resolution reaches 2.1 Å in the core regions of RNAP while peripheral regions mostly have ~3.0–4.0 Å resolution.

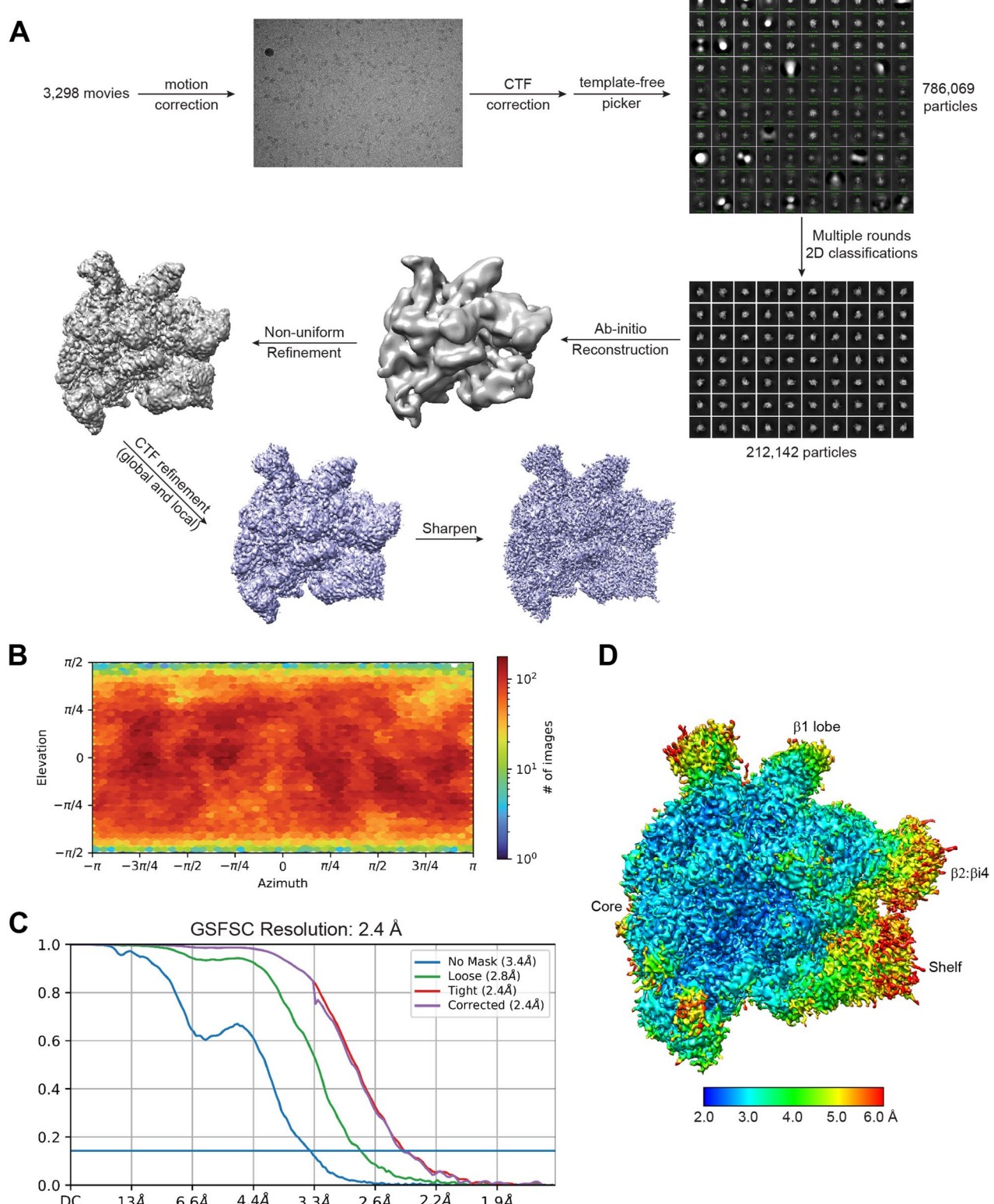

**Extended Data Fig. 2 | Cryo-EM workflow and resolution analysis for EC. a**, Processing and classification flow chart. **b**, Angular distribution of particle orientation. **c**, Fourier shell correlation (FSC) plot for half-maps with 0.143 FSC criteria indicated. The nominal resolution is determined to be 2.4 Å. **d**, A cryo-EM map colored according to local resolution. The resolution reaches 2.4 Å in the core regions of RNAP while peripheral regions mostly have ~3.5–5.0 Å resolution.

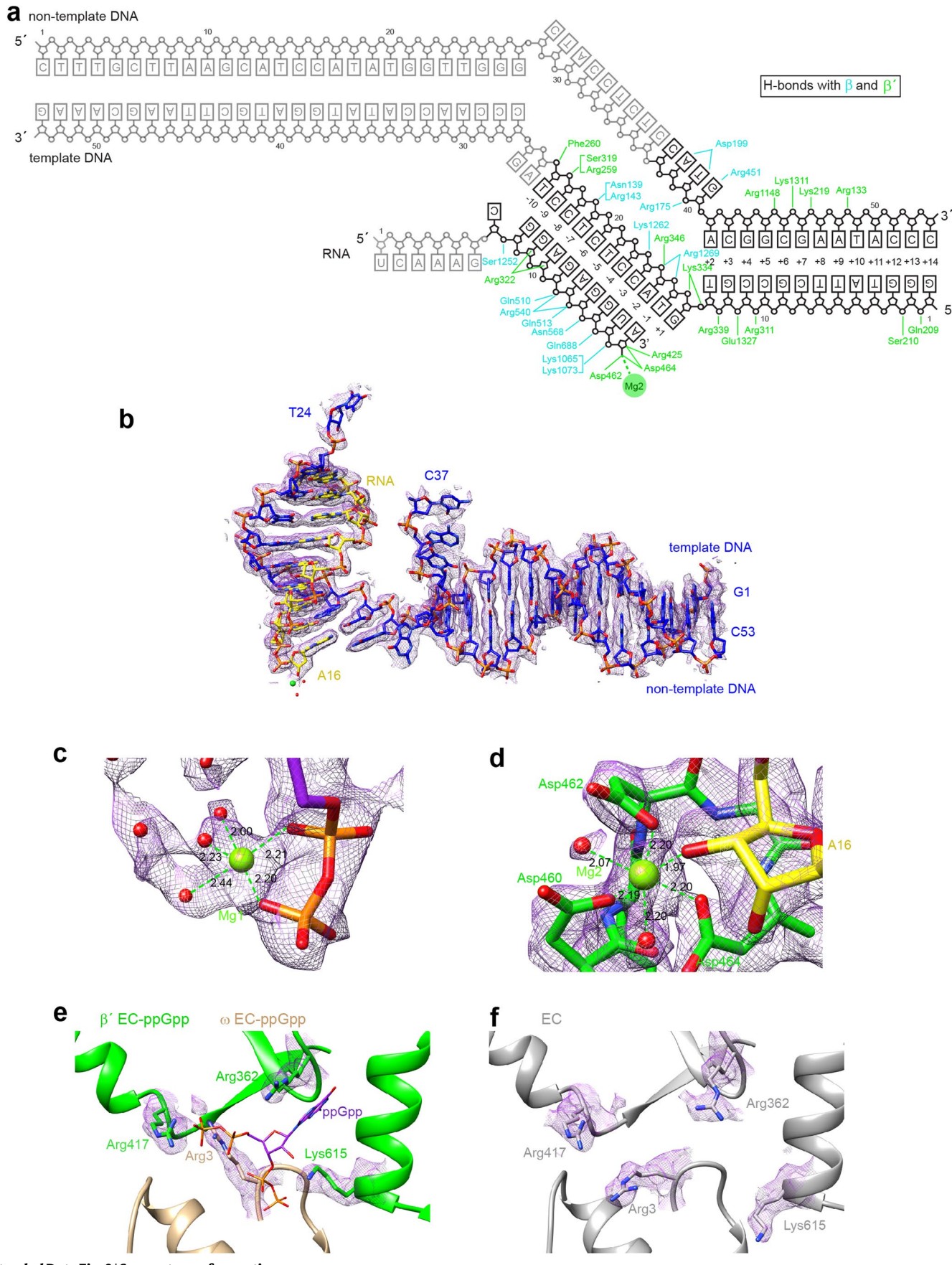

**Extended Data Fig. 3 | See next page for caption.**

**Extended Data Fig. 3 | Representative cryo-EM maps of the ppGpp-bound EC. a**, Schematic of the nucleic acid scaffold used in the study and observed interactions with RNAP. Parts of the scaffold visible and invisible in the cryo-EM maps are in black and gray color, respectively. The same nucleotides are visible in both EC and EC-ppGpp maps. Amino acids potentially hydrogen bonded with nucleic acids originate from β (cyan) and β´ (green) subunits of EC-ppGpp. Most interactions are preserved in EC. Only several residues form new interactions (β′ Lys216 and β′ Lys213), loose interactions (β′ Asn209 and β′ Arg259) or weaken bonding (β Asp199 and β Q513) predominantly because of small shifts or alternative rotamer conformations in the not well-resolved maps. **b**, Cryo-EM map (mesh) of the scaffold shown with the final refined EC-ppGpp structure. **c**, **d**, Zoomed-in images highlighting quality of the cryo-EM map. The views show the ppGpp-bound $Mg^{2+}$ cation (green sphere) and surrounding water molecules (red spheres) in panel (c) and the cryo-EM map around the $Mg^{2+}$ cation in the catalytic site in panel (d). **e**, **f**, Zoomed-in views of the ppGpp-binding site in the EC-ppGpp (e) and EC (f) reconstructions that highlight maps corresponding to different positions of the amino acid side chains.

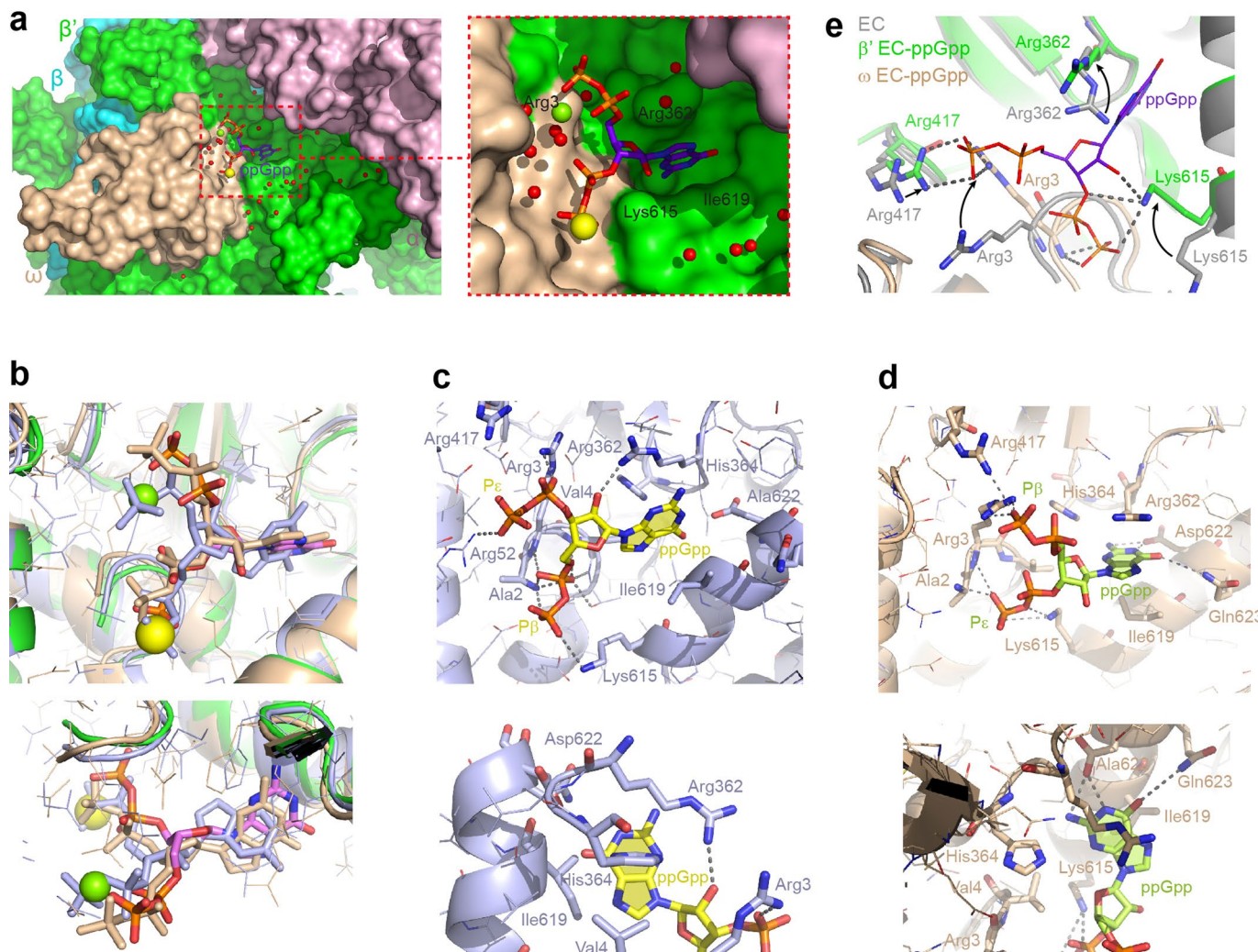

**Extended Data Fig. 4 | Details of ppGpp binding to Site 1 in *E. coli* transcription elongation complex. a**, Binding of ppGpp in a crevice between α and β′/ω subunits of RNAP. Proteins are in surface representation. The $Mg^{2+}$ cation, $Na^{+}$ cation, and water molecules are depicted as green, yellow, and red spheres, respectively. Note that the spheres are not intended to represent the actual sizes of these ions and molecules. **b**, Superposition of ppGpp binding sites from the current work (in colors) with the structures of the transcription initiation complexes (PDB ID code 4JKR, light blue, and 7KHI, beige). All-atom superposition was centered on amino acids β′ 617–635 and ω 45–56. Top and

bottom panels show side and top views, respectively. **c**, Binding of ppGpp in the 4.2 Å 4JKR structure. Putative hydrogen bonds (<4.0 Å) are shown in dashed gray lines. Amino acids engaged in ppGpp binding in the ppGpp-EC structure are shown in sticks. **d**, Binding of ppGpp in the 3.62 Å 7KHI structure. **e**, Conformational adjustments, shown by arrows, of amino acid side chains upon ppGpp binding. Moving moieties are in sticks. ppGpp is in thin sticks. RNAPs were superposed on the core. The structure of EC is in gray color, the structure of EC-ppGpp is in green (β′) and light brown (ω).

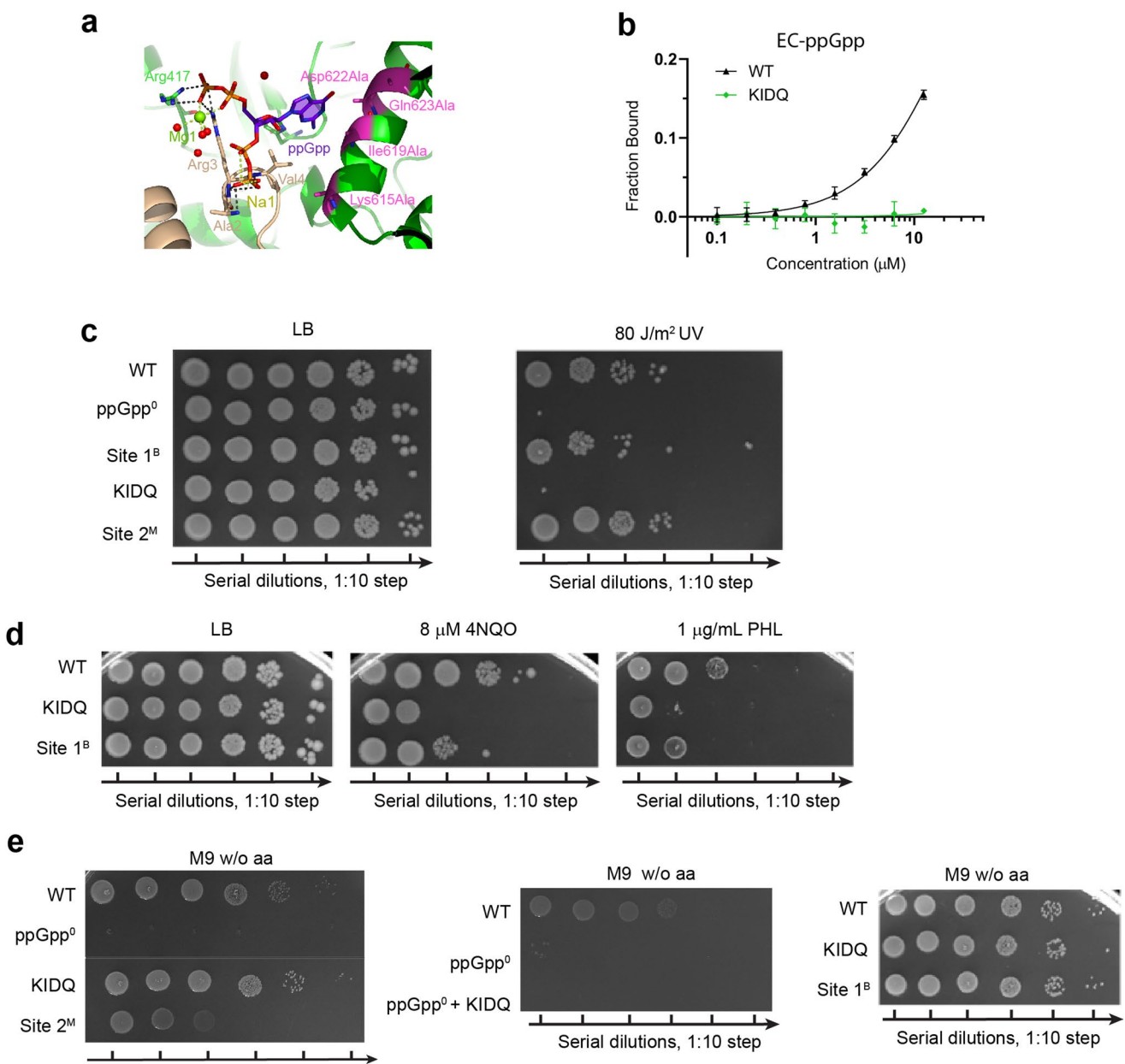

**Extended Data Fig. 5 | Mutations of RNAP and their effects. a**, The KIDQ mutation projected on the structure of the ppGpp-bound EC. All mutated amino acids (magenta) were replaced by alanines, resulting in loss of hydrogen bonding and other interactions with the nucleobase and backbone of ppGpp. **b**, Binding affinity measurements using radiolabeled ppGpp with wild type and mutant EC. Data points are mean ± SDs in bars (number of independent biological experiments n = 3). WT EC data were fit using single binding site model (GraphPad Prism), resulting in $K_D \approx 17\,\mu M$ and $B_{max} \approx 0.4$. Binding affinity for the KIDQ mutant cannot be reliably estimated and can reach high micromolar or low millimolar values. **c**, Effect of UV exposure at 80 J/m² on viability of WT and mutated *E. coli* cells. Bacterial cultures were irradiated by UV, serially diluted, and plated on an LB agar plate. Experiments in panels c-e were independently repeated 5 times with similar results. **d**, Comparison of the effects of genotoxic agents on Site 1ᴮ and Site 1 KIDQ mutants. Bacterial strains were grown in the presence of 4-nitroquinoline 1-oxide (4NQO) or phleomycin (PHL). **e**, Growth of WT and mutant *E. coli* cells on minimal media without amino acids.

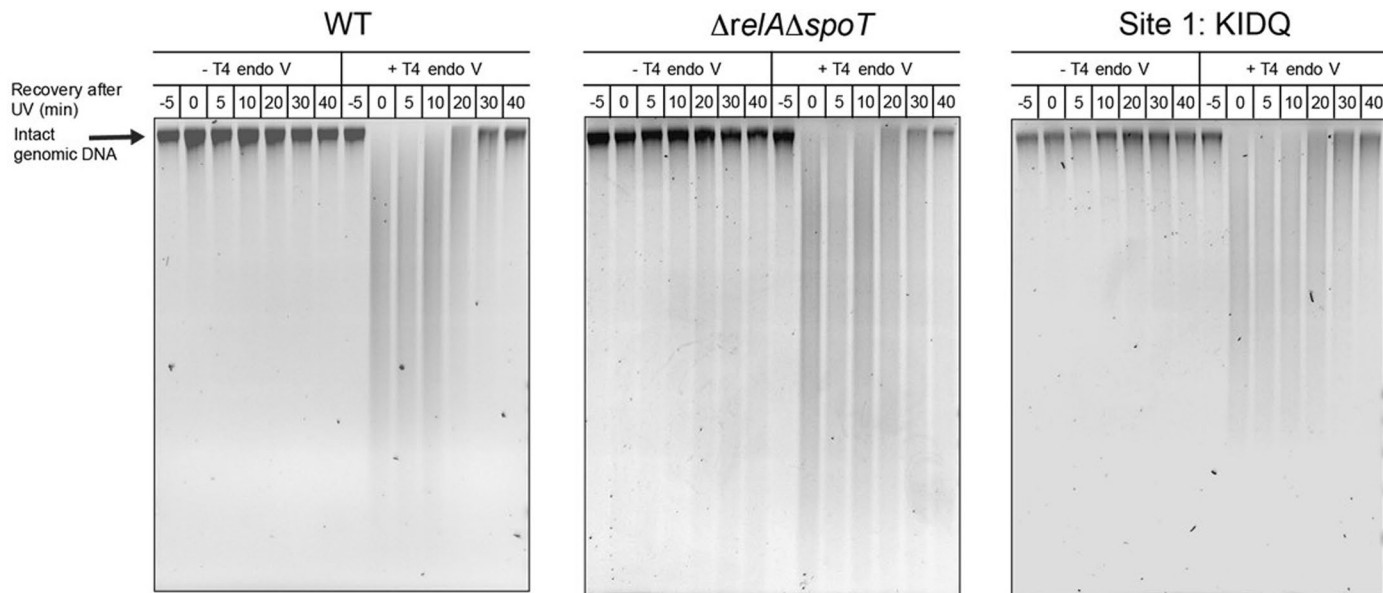

**Extended Data Fig. 6 | Representative genome-wide DNA repair experiments.** Wild-type MG1655 and mutant *E. coli* cells were UV irradiated and allowed to recover in dark. At the indicated times, the genomic DNA was purified and treated with T4 endonuclease V to convert UV-induced lesions (CPDs) into DNA breaks. The resulting DNA samples were analyzed on 0.5% alkali-agarose gels and the bands corresponding to the intact genomic DNA were quantified as described in Fig. 2e.

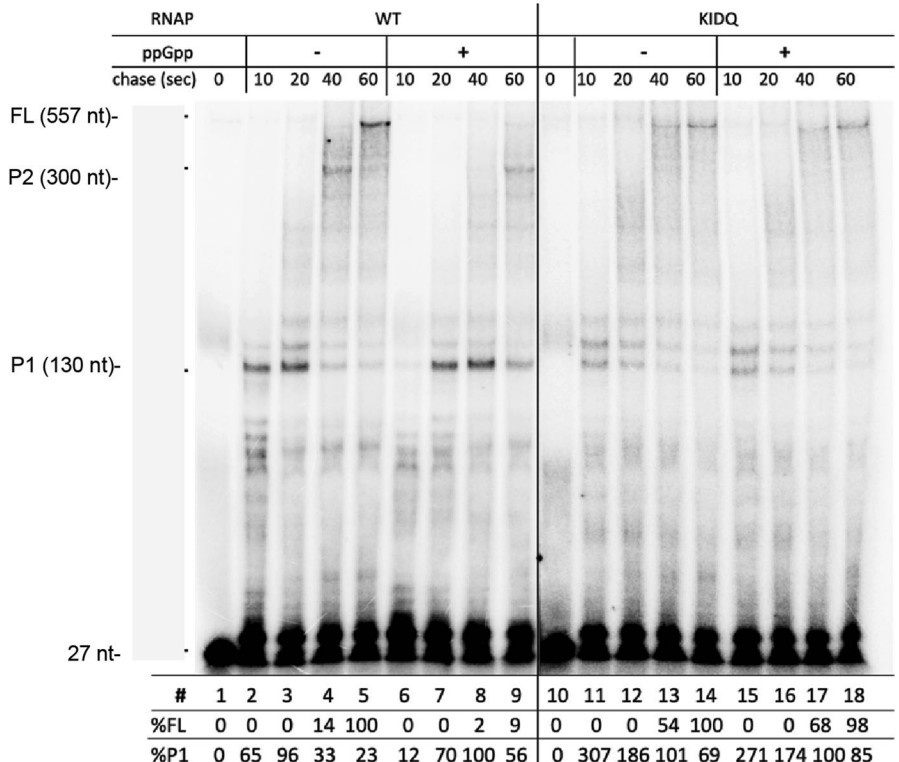

**Extended Data Fig. 7 | Effect of ppGpp in the presence of DksA and NusG on transcription elongation.** Kinetics of a single-round transcription with WT (left) and mutant (right) RNAPs in the absence and presence of ppGpp analyzed by PAGE. DksA (1 µM) and NusG (1 µM) were present in all reactions. FL, full-length RNA; P1 and P2, major transcriptional pauses. Bottom: data for FL and P1 were normalized to the corresponding bands in lanes 5 and 14 (FL), and lanes 8 and 17 (P1). Values were obtained in 2 independent experiments with an error margin of less than 15%.

**Extended Data Table 1 | Bacterial strains and plasmids used in the current study**[75,85,86]

| Strain | Description | Source |
|---|---|---|
| *Escherichia coli* CF12257 (ppGpp$^0$) | *relA256 spoT212* | Dr. Cashel[85] |
| *Escherichia coli* MG1655 | F$^-$ λ$^-$ rph-1 | Laboratory stock collection |
| *Escherichia coli* SP1231 | MG1655 *rpoC* K615A I619A D622A Q623A | This work |
| *Escherichia coli* SP0774 | MG1655 *rpoC* N680A K681A | This work |
| *Escherichia coli* SP0772 | MG1655 *rpoC* R362A R417A K615A *rpoZ* (Δ2-5) | This work |
| *Escherichia coli* SP1420 | *relA256 spoT212 rpoC* K615A I619A D622A Q623A | This work |
| *Escherichia coli* SP1636 | MG1655 *rpoC* K615A I619A D622A Q623A N680A K681A | This work |
| **Streptomyces morookaensis CF16775** | Produces extract with (p)ppNpp synthetic activity | Dr. Cashel[75] |
| **Plasmid** | Description | Source |
| **pVS10** | Wild-type *E. coli* RNAP overexpression plasmid | ref. 86 |
| **pSP-KIDQ** | RNAP β′ K615A I619A D622A Q623A mutant overexpression plasmid | This work |

**Extended Data Table 2 | Oligonucleotides used in the study**

| Oligonucleotide | Sequence | Description |
|---|---|---|
| **oA1** | TGACACGGAACAACGGCAAACACG | 16S rRNA leader, RT-qPCR |
| **oA2** | TGCATAATACGCCTTCCCGCTACA | 16S rRNA leader, RT-qPCR, primer extension |
| **o16S-3** | GTTACCGTTCGACTTGCATGT | 5′-mature 16S rRNA, primer extension |
| **oS1** | GAGCAAGCGGACCTCATAAA | Stable 16S rRNA, RT-qPCR |
| **oS2** | GGCATTCTGATCCACGATTACTA | Stable 16S rRNA, RT-qPCR |
| **oIra1** | CTGAGTTGTTATTTAAGCTTGCC | *iraP*, RT-qPCR |
| **oIra2** | AAGCATTGCAGTGACGATAA | *iraP*, RT-qPCR |
| **ntDNA** | CTTTGCTTAAGCATCCATATGGTTGG GCTACCTCTCCATGACGGCGAATACCC | Non-template DNA strand |
| **tDNA** | GGGTATTCGCCGTGTACCTCTCCTAGCC CAACCATATGGATGCTTAAGCAAAG | Template DNA strand |
| **oRNA** | rUrCrArArArGrCrGrGrArGrArGrGrUrA | Mimic of newly synthesized RNA |
| **sLacZ1** | GTGGAATTGTGAGCGGATAAC | Short *lacZ* amplicon |
| **sLacZ2** | GCTGCAAGGCGATTAAGTTG | Short *lacZ* amplicon |
| **lLacZ1** | GTGGAATTGTGAGCGGATAAC | Long *lacZ* amplicon |
| **lLacZ2** | CCATGACCTGACCATGCAGAGGATG | Long *lacZ* amplicon |

Alexander Serganov

# Reporting Summary

## Statistics

For all statistical analyses, confirm that the following items are present in the figure legend, table legend, main text, or Methods section.

| n/a | Confirmed | |
|---|---|---|
| ☐ | ☒ | The exact sample size ($n$) for each experimental group/condition, given as a discrete number and unit of measurement |
| ☐ | ☒ | A statement on whether measurements were taken from distinct samples or whether the same sample was measured repeatedly |
| ☐ | ☒ | The statistical test(s) used AND whether they are one- or two-sided<br>*Only common tests should be described solely by name; describe more complex techniques in the Methods section.* |
| ☒ | ☐ | A description of all covariates tested |
| ☐ | ☒ | A description of any assumptions or corrections, such as tests of normality and adjustment for multiple comparisons |
| ☐ | ☒ | A full description of the statistical parameters including central tendency (e.g. means) or other basic estimates (e.g. regression coefficient) AND variation (e.g. standard deviation) or associated estimates of uncertainty (e.g. confidence intervals) |
| ☐ | ☒ | For null hypothesis testing, the test statistic (e.g. $F$, $t$, $r$) with confidence intervals, effect sizes, degrees of freedom and $P$ value noted<br>*Give P values as exact values whenever suitable.* |
| ☒ | ☐ | For Bayesian analysis, information on the choice of priors and Markov chain Monte Carlo settings |
| ☒ | ☐ | For hierarchical and complex designs, identification of the appropriate level for tests and full reporting of outcomes |
| ☒ | ☐ | Estimates of effect sizes (e.g. Cohen's $d$, Pearson's $r$), indicating how they were calculated |

*Our web collection on statistics for biologists contains articles on many of the points above.*

## Software and code

Policy information about availability of computer code

| Data collection | CryoEM: Leginon 3.5 |
|---|---|
| Data analysis | CryoSPARC v 3.3.2<br>Phenix v 1.17.1<br>Coot v 0.8.9.2<br>PyMol v 2.5.4<br>Chimera v 1.15<br>GraphPad Prism v 8.4.3 |

For manuscripts utilizing custom algorithms or software that are central to the research but not yet described in published literature, software must be made available to editors and reviewers. We strongly encourage code deposition in a community repository (e.g. GitHub). See the Nature Portfolio guidelines for submitting code & software for further information.

## Data

Policy information about availability of data

All manuscripts must include a data availability statement. This statement should provide the following information, where applicable:

- Accession codes, unique identifiers, or web links for publicly available datasets
- A description of any restrictions on data availability
- For clinical datasets or third party data, please ensure that the statement adheres to our policy

CryoEM maps and atomic coordinates are deposited in the Electron Microscopy Data Bank and Protein Data Bank, respectively. Accession codes are PDB 8FVR and

# Field-specific reporting

Please select the one below that is the best fit for your research. If you are not sure, read the appropriate sections before making your selection.

☒ Life sciences ☐ Behavioural & social sciences ☐ Ecological, evolutionary & environmental sciences

For a reference copy of the document with all sections, see nature.com/documents/nr-reporting-summary-flat.pdf

# Life sciences study design

All studies must disclose on these points even when the disclosure is negative.

| | |
|---|---|
| Sample size | No sample size calculation was performed. Independent experiments were conducted for reproducibility. The number of experiments were chosen based on the standards of the field (at least 2 biological replicates) and the total number of individual experiments depended on the reproducibility of the results. |
| Data exclusions | Particle images were excluded during cryoEM data processing following 2D classification based on lack of high-resolution detail following standard procedure for cryoEM structure determination. |
| Replication | Replication of experiments was successful. The number of independent biological experiments is indicated in Methods and Figure legends. |
| Randomization | Randomization was not used for this study as outcomes did not depend upon researcher bias. |
| Blinding | Blinding was not employed for this study as outcomes did not depend upon researcher bias. Replication of experiments showed good reproducibility. |

# Reporting for specific materials, systems and methods

We require information from authors about some types of materials, experimental systems and methods used in many studies. Here, indicate whether each material, system or method listed is relevant to your study. If you are not sure if a list item applies to your research, read the appropriate section before selecting a response.

## Materials & experimental systems

| n/a | Involved in the study |
|---|---|
| ☒ ☐ | Antibodies |
| ☒ ☐ | Eukaryotic cell lines |
| ☒ ☐ | Palaeontology and archaeology |
| ☒ ☐ | Animals and other organisms |
| ☒ ☐ | Human research participants |
| ☒ ☐ | Clinical data |
| ☒ ☐ | Dual use research of concern |

## Methods

| n/a | Involved in the study |
|---|---|
| ☒ ☐ | ChIP-seq |
| ☒ ☐ | Flow cytometry |
| ☒ ☐ | MRI-based neuroimaging |

