## [Peer Review File · Nature Structural & Molecular Biology]

Peer Review Information

Manuscript Title: Control of Transcription Elongation and DNA Repair by Alarmone ppGpp

Corresponding author name(s): Evgeny Nudler, Alexander Serganov

Reviewer Comments & Decisions:

Decision Letter, initial version:

Message:

Dear Dr. Nudler,

Thank you for submitting your manuscript "Allosteric Control of Transcription Elongation and DNA Repair by Alarmone ppGpp". I apologize for the delay in processing your manuscript, which resulted from difficulties in obtaining referees' reports. Nevertheless, we have received comments from the 3 reviewers who have evaluated your manuscript are below. Unfortunately, after carefully considering their comments, we cannot offer to publish your manuscript in Nature Structural & Molecular Biology.

You will see that while the referees find the work of potentially interesting, they raise serious technical concerns regarding the cryo-EM analysis, in terms of sample preparation and lack of density of the upstream and downstream duplex, as well as issues with the interpretation and consistency from the in vitro and bacterial studies.

However, if further experimentation, analysis, and revisions allow you to address the referees concerns in full, we would be prepared to consider an appeal of our decision, on the condition that no related work is published in the interim or has been accepted in our journal. Please contact me to discuss an appeal and potential revision. Please note that, until we have the opportunity to read the revised manuscript in its entirety, we cannot promise that it will be sent back for peer review.

I am sorry we could not be more positive on this occasion. I hope that you find the referees' comments useful in deciding how best to proceed.

Sincerely,

Carolina

Carolina Perdigoto, PhD
Chief Editor
Nature Structural & Molecular Biology
orcid.org/0000-0002-5783-7106

Author Rebuttal to Initial comments

Response to reviewers

We thank the reviewers for their constructive feedback on our manuscript. The reviewers' comments and suggestions have enabled us to make data analysis and experimental revisions that clarify and strengthen our findings. As described below, we have revised the manuscript to address each of the points raised by the reviewers.

Reviewer #1:

Remarks to the Author:

The manuscript entitled "Allosteric control of Transcription elongation and DNA repair by alarmone ppGpp by Weaver et al. reported the structural, biochemical, and genetic evidence that ppGpp allosterically controls the E. coli RNA polymerase (RNAP) during transcription elongation via a specific site that is nonfunctional during initiation. Detailed structural analysis showed that how ppGpp alters E. coli RNAP conformational dynamics to make it backtracking inclined state. Combining structural and functional genomics approaches, authors convinced that how the two distinct ppGpp-binding sites on RNAP, Sites 1 and 2, control transcription elongation and initiation, respectively. They further emphasized that ppGpp binding to RNAP Site 1 during elongation is important for promoting DNA repair, and suggested that Site 1 could be an attractive new target for rational design of small molecule modulators of transcription with potential antimicrobial agents.

There are several issues in the cryo-EM structure determination and in the planning of the transcription experiments as listed below. These issues must be addressed to justify authors claim about the ppGpp binding at the site 1 for regulating RNAP transcription elongation before considering publishing this work in the high standard NSMB journal.

We appreciate the reviewer's thoughtful comments, which helped to improve our manuscript substantially. We have addressed all the reviewer's concerns as outlined below.

1. CryoEM structure determinations of EC in the absence and in the presence of ppGpp.

I appreciated authors to provide the cryo-EM density maps and coordinates of for checking the qualities of maps and final structures. I calculated difference maps (complex minus RNAP) showing the densities corresponding the DNA/RNA in these complexes (see a figure below). There are several issues in the modeling of DNA/RNA.

- In EC, density of the upstream DNA is very weak and because of this, the upstream DNA is not modeled as B-form DNA. I recommend deleting the upstream DNA model or refine it with the DNA secondary structure restraint (base pairing and base stacking).

We have determined new cryo-EM structures of EC-ppGpp and EC at 2.1 and 2.4 Å resolution, respectively using longer DNA template with a different transcription bubble, as recommended by Reviewer 3 (**new Figs.1 and Extended Data Fig. 3a**). In these structures, we retained nucleotide residues, which have clear maps. We prepared new figure, where we show visible parts of the nucleic acid scaffold in schematic (**new Extended Data Fig. 3a**) and with maps (**new Extended Data Fig. 3b**). The upstream portion of the DNA scaffold has poor map and was not modelled.

- In EC-ppGpp, there is no density corresponding to the downstream DNA (can see only four bases of the template DNA but not density for non-template DNA). I recommend deleting the downstream DNA model.

Authors must document differences in the DNA densities of these complexes and link it to the effect from the ppGpp binding on RNAP. It seems like RNAP decreases its grip of DNA (both upstream and downstream) upon binding ppGpp.

In the new high-resolution structures of EC and EC-ppGpp, we see the same parts of the scaffold and almost same patterns of DNA recognition. As mentioned above, we documented these features in **new Extended Data Fig. 3**.

In addition, there is a density at the substrate binding site in the EC (indicated by an arrow), suggesting that the RNA in the EC is in equilibrium between pre- and post-translocation states. There is no density at the substrate binding site in the EC-ppGpp, indicating that the RNA is in the post-translocation state. This observation is not consistent with the observations where the ppGpp binding makes RNAP prone to backtracking. Authors must discuss the discrepancy between the structural and biochemical observations regarding the ppGpp binding effect on RNAP backtracking.

In the new higher resolution structures, the RNA is in the post-translocation state. As pointed out by Reviewer 3, our scaffold would not allow backtracking because of the mismatches in the transcription bubble. The purpose of the structural work was to establish the exact position of ppGpp in the EC, not to confirm the backtracking state it promotes. The ability of ppGpp to do so has been shown in our previous work in vitro and in vivo (PMID: 27199428).

2. Mg coordination by ppGpp phosphate groups.

Authors assigned densities found near 5'-phosphate group of ppGpp as Mg²⁺ and water molecule. However, the Mg has only three coordinating sites (two non-bridging oxygens and water), which is uncommon. Most Mg²⁺ has hexa coordination. The density may be Na⁺. See a reference below. (Batra, V. K. et al. Magnesium-Induced Assembly of a Complete DNA Polymerase Catalytic Complex. *Structure* 14, 757–766 (2006).)

The reviewer is correct in stating that most Mg²⁺ cations have hexa coordination while coordination of Na⁺ is more relaxed. Geometry of the coordinated Mg²⁺ cations is typically octahedral with coordination bonds shorter (~2.0-2.2 Å) than those formed by Na⁺ (~2.3 Å). At the ~2.5 Å resolution of the original structure, it was impossible to distinguish between the two cations, and Mg²⁺ was assigned as a cation with a higher affinity for phosphates. In the higher resolution structures presented in the revised manuscript, the sharpened map revealed one penta-coordinated Mg²⁺ cation forming octahedral coordination geometry with two non-bridging oxygen atoms and three water molecules. There is no map for the fourth water molecule, and it was not added to the structure. The sharpened map for the second metal cation bound to ppGpp does not clearly reveal octahedral geometry and we relied on the unsharpened map in positioning the cation at the ~2.5 Å distance from a non-bridging oxygen atom. Because of this distance, we assigned this map peak to Na⁺ cation and refrained from adding coordinated water molecules. Please note that these assignments of metal cations have been confirmed using a 2.3 Å resolution map generated from the independently collected data set. Maps for metal cations are now shown in **new Extended Data Fig. 3c,d**.

3. In vitro transcription assay

1) To investigate the effect of ppGpp on transcription elongation, authors compared the RNA elongation and pause escape of both wild-type and ppGpp-binding defect KIDQ mutant with and without ppGpp condition. Authors described that the KIDQ mutant is no longer response to the ppGpp. However, KIDQ mutation by itself influences RNAP transcription, such as less pausing at P1 position and faster transcription compare with wild-type RNAP. Authors must discuss the effect of KIDQ mutation on RNAP transcription.

Indeed, KIDQ mutant is slightly faster in vitro than the wt RNAP. We believe that this reflects the nature of Site 1 as the highly responsive pocket that enables the allosteric control of RNAP by a small ligand, such as ppGpp. It is, therefore, not surprising that the same mutations that prevent ppGpp binding would also alter the properties of the enzyme. However, we also note that KIDQ mutation has no obvious growth defects (**new Figs.2a and 3a**), no effect on stringent response (revised **Fig.3c, d and new e**), and fully epistatic with ppGpp(0) mutation with respect to genotoxic stress (**revised Fig. 2b**), arguing that KIDQ phenotype in vivo is primarily accounted for its inability to bind ppGpp and to control RNAP elongation via Site 1. As requested, we now better discuss this point in the revised manuscript.

In addition, authors tested RNAP backtracking at ppGpp-sensitive pausing site (Fig. 4b). Since P1 pause half-life is long enough even without ppGpp (lane 3 in Fig. 4a), authors must carry out

the same experiment (Fig. 4b) in the absence of ppGpp and compare the results (with or without ppGpp).

As requested, we now also present the results of the experiment of Fig. 4b in the absence of ppGpp. By comparing the two parts, with and without ppGpp, one can conclude that ppGpp strengthens the backtracking-prone (GreB-sensitive) pauses (P1 and P2), as no readthrough transcripts can be visible above P2 in the ppGpp case. The **revised Fig.4b** is shown below.

4) ppGpp concentration is changed during stringent response whereas the DksA concentration is constant. The ppGpp-binding site 2 is depending on DksA. DksA can influence the transcription not only at the initiation stage but also during elongation as reported by the Zenkin/Yuzenkova group (Roghalian, M., Zenkin, N. & Yuzenkova, Y. Bacterial global regulators DksA/ppGpp increase fidelity of transcription. *Nucleic Acids Res* 43, (2015).). Furthermore, most elongating RNAP associate with elongation factor NusG. To investigate the *E. coli* RNAP transcription near physiological condition, authors should repeat the in vitro transcription (Fig 4a) in the presence of DksA and NusG to evaluate the ppGpp effect.

This is a good point, which we addressed experimentally, as requested by the reviewer. We repeated the experiment of Fig. 4a in the presence of NusG and DksA (**new Extended Data Fig. 6; see also below**). While the presence of DksA and NusG accelerated the overall elongation rate with and without ppGpp, these factors do not change the principle results we reported in Fig. 4a, i.e. KIDQ mutation completely eliminated the ability of RNAP to respond to ppGpp during elongation.

Minor points:

1. Fig 2a: pattern of bar graph in panel 2 and 3; nice to have similar pattern as other bar graphs.

The appearance (colors, etc) are now the same between the graphs. Please note that the bar graphs in panel A have a logarithmic Y axis while other graphs of Fig. 2 have a linear Y axis. Therefore, we retained guiding lines to emphasize this difference.

2. Proofread for some typo and same representation pattern of same words throughout manuscript.

Done

3. In extended discussion, “Comparison with 6WRD and 6WRG: These unpublished 3.62 Å and 3.58 Å cryo-EM structures contain the RNAP-RpoD-DksA-ppGpp complex initiating transcription on the *rrnBP1* promoter.” These structures were published in Shin et al., Nature Communications, 2021 (DOI: 10.1038/s41467-020-20776-y) as PDB 7KHE and 7KHI. Cite this paper in page 3 “Interactions between ppGpp and RNAP in Site 1 are similar in the ppGpp-EC, ppGpp-IC, and ppGpp-IC-DksA structures 15,16 albeit, with some notable differences (Extended Data Figs. 3e-g and Note S1).”

We have updated the manuscript and cited Shin et al, as suggested by the reviewer.

4. Page 4 “Together, these data show that Site 1 does not contribute to the stringent response upon amino acid starvation, while Site 2 is critical for the response as previously reported.” In Fig. 2g, KIDQ mutant strain expresses the rRNA 2 times more compare with the wild-type strain, suggesting the effect on site 1 during the stringent response.

We are not sure we fully understand this point. KIDQ generates practically the same amount of rRNA as the WT (compare lanes 1 and 3 of the Primer Extension assay in new Fig. 3c) and responded to SHX the same way as the WT (compare lanes 2 and 4). Fig. 3d (RT-qPCR) is in a good agreement with this result and although the average values of repression are not identical in WT and KIDQ, they are qualitatively the same considering the error margin in the KIDQ experiment. The principal feature of the “stringent response” is the *inhibition* of rRNA expression in response to amino acid starvation (in this case the treatment with serine hydroxamate). In contrast, Site 2 mutant exhibits the same elevated expression of rRNA as does ppGpp(0) and fails to respond to SHX.

5. Final paragraph of Discussion (page 6). “The allosteric mechanism of regulation by ppGpp described here for *E. coli* must be conserved among bacteria”. This statement is not true since ppGpp binding site 1 is conserved in proteobacteria (see Fig 3 in Hauryliuk, V., et al., Nat Rev Microbiol., 2015) and the site 2 is DksA-dependent and DksA is also conserved only in proteobacteria (see Ross et al., Mol Cell, 2016).

We agree and have corrected the corresponding sentence in the Discussion.

Reviewer #2:

Remarks to the Author:

Weaver et al. sought to elucidate ppGpp’s role in transcription-coupled repair of DNA. They present a high resolution Cryo-EM structure of an RNAP elongation complex, bound to ppGpp, and characterized the conformational changes that occur upon binding. Based on their experimental set-up, they found that ppGpp could bind to elongating RNAP in the so-called “site 1”, and this caused a slight but stable shift in its swivel module. The authors designed a new site 1 ppGpp-insensitive mutant, designated KIDQ, to further study this site’s significance in RNAP’s established roles in response to amino acid starvation and DNA-damaging agents. The authors argue for site 1 having a role in transcription-coupled repair (TCR) based on spot plating, monitoring of region-specific DNA lesions by RT-PCR, and also monitoring genome-wide DNA repair. The authors also show evidence that site 2 plays a larger role in the stringent response than site 1, based on spot plating, growth curves, and RT-qPCR of ribosomal RNA. Lastly, the authors performed in vitro transcription with their new KIDQ mutant and it appears to be insensitive to ppGpp at pauses. Notably, the ppGpp-sensitive pause was prone to backtracking. Overall, I think this is a lucidly written manuscript that demonstrates the role in and mechanism of ppGpp binding to RNAP Site 1 regulating nucleotide excision repair by promoting transcription-coupled repair, but would benefit from additional supporting evidence for clarification.

We thank the reviewer for their overall enthusiasm with our manuscript. We found the reviewer's suggestions to be very helpful in clarifying our message and highlighting useful experiments to conduct. These changes and suggestions have been incorporated into the manuscript and detailed below in response to the individual comments.

Major comments:

1. The authors rely on a new site 1 ppGpp-null mutant which they call KIDQ. While it is true that these point mutations abolish the site 1 interaction with ppGpp, the authors also showed that *in vitro* this mutant is faster at elongation and is less responsive to pauses. Can the authors provide insights and data as to how the KIDQ mutations affect the RNAP EC structure? Also, this could easily have unknown consequences *in vivo* which the authors have not accounted for, since pausing can be an important precursor to TCR. If the authors cannot demonstrate this mutant has no ppGpp-independent effects *in vivo*, they may consider repeating experiments (treatments with spot plating, repair efficiency, genome-wide DNA repair experiments) using the original site 1 mutant which is still sensitive to pauses. It may also require more justification as to why this mutant is “optimized” vs. the original site 1 mutant – more than just the fact that the latter was based on low-resolution structures.

Indeed, KIDQ RNAP is mildly faster *in vitro*. We believe that this reflects the nature of Site 1 as a highly responsive pocket enabling the allosteric control of RNAP by a small ligand, such as ppGpp. It is, therefore, not surprising that the same mutations that prevent ppGpp binding also modulate the properties of the enzyme. However, we also note that the KIDQ mutation has no obvious growth defects (Figs.2a and 3a), no negative effect on stringent response (same as the WT) revised Fig.3c, d and new e), and fully epistatic with the ppGpp(0) mutation (**revised Fig. 2b**), demonstrating that KIDQ phenotype *in vivo* primarily accounts for its inability to bind ppGpp and to control RNAP elongation and TCR by ppGpp via Site 1. We now better discuss this point in the revised manuscript.

We now better explain the difference between “old” site 1 mutant (b' R362A R417A K615A D2-5w), which did not affect the interaction with ppGpp nucleobase, and KIDQ. KIDQ eliminates critical ppGpp-protein interactions that recognize the nucleobase, sugar, and 3' phosphates of ppGpp, which remain in the “old” site 1 mutant.

As requested by the reviewer, we prepared the “old” site 1 mutant and conducted the experiments with it to compare the effects with the KIDQ mutant. The “old” site 1 mutant has a milder defect in TCR (**new Extended Data Fig. 4c and d; see below**), as compared to KIDQ, which has the same strong negative effect as ppGpp(0). As mentioned above, this phenotype of KIDQ is fully epistatic with ppGpp(0) (**Fig. 2b**). Therefore, regardless of whether the old site 1 mutant were faster than the wt (we didn't examine its properties *in vitro*), it does not mimic the ppGpp(0) phenotype with respect to genotoxic stress, whereas KIDQ does.

2. In the section ‘Site 1 is critical for ppGpp-mediated nucleotide excision repair’, the authors state that, “Under DNA damage conditions, this mutant [KIDQ] displayed profoundly reduced survival, comparable with that of the ppGpp0 ($\Delta relA \Delta spoT$) strain (Fig. 2a and Extended Data Fig. 4c).” (Top of page 4) In figure 2a, the ppGpp0 mutant has a substantially lower % Cell count than the KIDQ mutant when subjected to 60 J/m² UV, while in extended data figure 4c, the ppGpp0 and KIDQ mutants have similar low viability (based on the representative plate; quantification not shown) when subjected to 80 J/m² UV. Is the difference between the two mutants in figure 2a statistically significant, and if so, why might the KIDQ mutant have higher viability under this condition?

The reviewer is right. Although KIDQ exhibits the same phenotype as ppGpp under higher UV doses and NFZ, we see some difference at the lower UV dose, which seems to be reproducible. As ppGpp has numerous cellular targets apart from RNAP (Wang et al., Nat Chem Biol 2019), we believe that some indirect effects of ppGpp can be responsible for those differences. We now mention this in the revised manuscript. However, most of the effect of ppGpp on TCR must be mediated by Site1 based on the overall compelling data we present in the manuscript.

3. In Fig. 2d, the authors demonstrate that on its own, the KIDQ mutant grows fine on minimal medium lacking amino acids, suggesting that it is dispensable for the stringent response, whereas the site 2 mutant grows poorly. I am curious whether the authors have tested the double mutant, site1-site2-, in these same conditions, since the two could have synergistic effects.

Following the reviewer’s suggestion, we constructed the double mutant, Site1(KIDQ)+Site2, and compared its growth on minimal media with Site1 and Site 2 mutants. The results are now

presented in **new Fig. 3a (see below)**. It is clear that Site1(KIDQ)+Site2 mutant behaves similar to Site 2 mutant, further supporting the notion that Site1 is dispensable for stringent response.

4. The authors show in Fig. 2f,g that KIDQ mutant responds similar to wildtype under stringent response induction by SHX in terms of rRNA transcript levels, but are the values statistically significantly different between WT and KIDQ, and P0 and Site 2? Do the authors have evidence for or against the KIDQ mutant's effects on transcription initiation/the structure of the initiation complex? If not, unknown effects of this mutant on the IC structure could impact this strain's ability to grow in these conditions by tuning initiation differently. Since the stringent response is also known to positively affect transcription at certain promoters, it could strengthen the argument to have similar measurements from a promoter like *iraP*.

The difference between WT and KIDQ in response to SHX was not significant, as was the difference between Site 2 and ppGpp(0). The difference between KIDQ and ppGpp(0) is significant. We now present these statistics for **Fig. 3d**.

In addition, we also now present *in vivo* data showing that KIDQ and WT RNAP initiate equally well at *iraP* promoter upon SHX treatment, whereas ppGpp(0) and Site2 mutants are defective in transcription activation from this promoter during stringent response. These results now added as **new Fig. 3e (see below)**.

5. The authors tested the kinetics of repair after exposure to DNA-damaging levels of UV, and found that both P0 and KIDQ strains were slower to repair the lesions. The timing of this

coincides with the previously published UV induction of ppGpp. The results suggest that site 1 has a role in TCR, but the authors did not present this set of experiments for either the site 2 or site1-site2- mutants, which would be important for claiming that site 1 is the major ppGpp binding site during elongation. Again, the KIDQ mutant in vitro was shown to be less sensitive to pauses, which could in turn decrease the TCR activity indirectly.

Site 2 mutant is as sensitive to DNA damaging agents and UV as the WT (Fig.2a). We, therefore, didn't expect it would have any effect on kinetics of CPD repair. However, following the reviewer suggestion, we analyzed the effect of Site 2 mutant on CPD repair within the lacZ locus (**revised Fig. 2e** see below). As expected, Site 2 mutant shows no difference in the rate of DNA repair comparing to the WT.

6. At the end of the discussion, the authors remark on the conservation of RNAP and that the mechanism they describe, “must be conserved among bacteria, as RNAP from evolutionary distant species, such as Mycobacteria, also responds to the alarmone during elongation.” It would be worth being less generalizing and more precise here (i.e., that this mechanism is likely conserved amongst proteobacteria as well as certain evolutionarily distant species). As noted in reference 21, RNAP site 1 is notably not conserved beyond proteobacteria. In the authors’ own paper (reference 47), ppGpp has a much weaker effect on Mycobacteria RNAP elongation.

We agreed and have modified our discussion accordingly.

Minor comments:

a. Typo in fig 3c y axis

Old Fig. 3 was removed.

b. Can the authors clarify the number of replicates for Fig 4? Statistics?

This information is now provided in the figure legend.

c. In Fig. 2a, the 1:10 dilutions on NFZ plates are interesting, because in the representative plate both P0 and KIDQ after the second dilution there seems to be less than 1/10 of bacteria that survive that dilution. It could be a function of the NFZ itself, but it is interesting to note, and could impact the cell counts.

We agree that this seems like an intrinsic property of NFZ at this concentration, as the effect is reproducible (see also Fig. 2b).

d. The authors prepared their cryo-EM samples in the absence of DksA, which can explain why ppGpp bound to site 1 and not site 2. But it doesn't rule site 2 out of having roles in elongation, especially with DksA being an elongation factor with potential roles in backtracking.

This is a valid point. To address it we repeated the experiment of Fig. 4a in the presence of DksA and NusG, as also suggested by Reviewer 1. The results are presented as **new Extended Data Fig. 7** (see also Reviewer 1 point 4). While the presence of DksA and NusG together accelerated the overall elongation rate with and without ppGpp, these factors did not change the principle results we reported in Fig. 4a, that is KIDQ mutation completely eliminates the ability of RNAP to respond to ppGpp during elongation, regardless of DksA.

Reviewer #3:

Remarks to the Author:

This manuscript by Weaver et al. investigates the impact of the alarmone ppGpp on RNA polymerase (RNAP) elongation complexes (EC). The manuscript describes a combination of in vitro, in vivo, and structural approaches to propose role of ppGpp in transcription-coupled DNA repair. I found the in vitro and in vivo experiments along with mutational studies convincing. The authors also determined two cryo-EM reconstructions of E. coli RNAP EC with and without ppGpp. Although the reconstructions reached high resolution and assigned unaccounted density to ppGpp, which agrees with previous findings, I have concerns about interpretation of the structural results and I also have questions about sample preparation and choice of nucleic acid substrates. The authors argue that ppGpp binding influences the conformational heterogeneity of the RNAP EC and narrows the range of RNAP swivel motions rendering it prone to backtracking. This is an interesting hypothesis but the observed differences are very subtle and thus I feel it is in the authors interest to respond to the following points and take them into account:

We appreciate the reviewer's scrupulous evaluation of our structural part of the paper and detailed and constructive comments. We tried our best to address all of them in full in the revised manuscript.

To this end, **we re-did the entire cryo-EM work**. As requested, (points 5 and 6), we prepared a longer nucleic acid scaffold, composed of two 53-nt DNA strands. When annealed, these strands form the 13-bp downstream DNA helix, a smaller 10-nt transcription bubble, and, a longer, 30-bp, upstream DNA helix. The scaffold schematic is now presented in **new Extended Data Fig. 3**. We collected a few data sets and achieved 2.1 Å resolution for the ECppGpp. To our knowledge, this is the highest resolution of any transcription complex reported so far. The new structure provides an unprecedentedly detailed information, highlighted by the observations of the ppGpp-bound metal cations and their coordinated water molecules (shown in **new Fig. 1 and Extended Data Fig. 3**). We can also see conformational adjustments in the amino acid side chains upon ppGpp binding to the polymerase.

This reviewer had concerns that the observed differences between EC and EC-ppGpp are subtle to justify the conclusion that ppGpp binding influences the conformational heterogeneity, arrived from 3D variability analysis, and narrows the range of RNAP swivel motion. Our hypothesis was supported by comparing more than one data set, with the highest resolution data sets reported in the original manuscript. Nevertheless, we share the concerns of the reviewer and have collected several more data sets for both EC-ppGpp and EC, with new scaffolds, at a much higher resolution than the original data. In these new data sets, 3D variability analysis did pick up differences between EC and EC-ppGpp, but the magnitude of the differences varied significantly across different pairs. Since, in theory, the particle distribution can be influenced by grid quality and particle picking and classification, we decided to remove 3D variability analysis until we better understand utility and limitations of this emerging technique. We have also decided to refrain from comparison with the published structures determined at a lower resolution.

1. The ppGpp-EC is more conformationally homogenous according to the authors – however, there were also twice as many particles to choose from. Could the authors just have isolated a more homogenous subset in the course of their classification? For example, the main class of the EC contains 403,000 particles (78% of the particles after 2D classification). The main class of the ppGpp-EC contains about 550,000 particles (47% of particles chosen after 2D classification) but a second class still contains almost 380,000 particles (32% of particles after 2D classification, extended fig. 1a) – what is the second class? I think a fairer comparison between the two datasets needs to include this second class (to reach a total of 79% of particles after 2D), then perform classification (the latest CryoSPARC supports 3D classification including masks for focused classification – see also points further down), then refine atomic models against different classes, then compare swivel module orientations of the extremes and/or particle numbers in different classes to see if there is really a difference.

Indeed, it is extremely difficult to prepare comparable grids of the same quality for EC and ECppGpp and maintain the same proportion of useful and junk particles. Therefore, in the revised manuscript, we decided not to use 3D variability analysis. The principal goal of the structural work in our manuscript was to precisely map the positioning of ppGpp in the transcription elongation complex and then validate the binding site *in vitro* and *in vivo*. All these goals have been fully accomplished.

2. Does the cross-correlation between map and model support the observed 0.3 degree difference. In other words, if the authors rigid-body dock the more swiveled RNAP into the less swiveled RNAP map (and vice versa), do they see significant differences in the cross-correlation between model and map. Phenix reports this on a per-residue basis, so one can specifically confirm this for the swivel module rather than globally (the latter may hide meaningful differences) and the authors should include this analysis to give the reader confidence that the differences are supported by the maps. I think the same should be done with models that had been fit to “extremes” in the 3D variability analysis (compare CC of swivel module of model A to map of extreme A with CC of swivel module in model A to map of extreme B and vice versa) and also after more rigorous 3D classification using all particles after 2D classification (see previous point).

Although new cryo-EM maps have a much higher resolution in the core, the peripheral regions and moving parts of RNAP have lower resolution and worse fit of the model into the maps. Therefore, in the revised manuscript, we refrained from extensive comparisons between the modules of EC and EC-ppGpp as well as comparisons with published structures. We did, however, observe a small but noticeable difference in the omega subunit, a region of reasonably high resolution, and also observed changes in the amino acid side chains upon ppGpp binding (**Extended Data Figs. 3e,f and 4e**). We are confident in these changes.

3. The authors compare their structures to two backtracked complexes (6RI9 and 6RIP) but those adopt a swiveled and a non-swiveled conformation, respectively. The authors should discuss and explain how they imagine that narrowing the range of swivel module orientations would favor backtracking given that a backtracked complex itself appears to swivel? Per that publication, the authors explain an equilibrium exists between the two but a backtracked conformation/sequence would shunt the equilibrium to a swiveled conformation over nonswiveled.

In the revised manuscript we removed the comparison and no longer discuss it.

4. I see not much difference in the ppGpp binding site between the two provided coordinate sets (EC and ppGpp-EC). Even most rotamers are the same and many “differences” could simply be explained by coordinate error. Even if we compare structures with much larger differences in their swivel module orientations (e.g. the authors compared to a swiveled hisPEC-NusA and a backtracked non-swiveled complex), there is very little structural difference in site 1 making it difficult for this reviewer to appreciate how ppGpp could influence the range of swivel module orientations when it binds to a pocket that does not change upon swiveling. The authors should elaborate their hypothesis and discuss this in the manuscript.

The cryo-EM structures represent static structures used primarily for visualizing ppGpp binding to EC. These structures may not report on the differences in motion, associated with the

catalytic site. Although we attempted to investigate the molecular mechanism for the ppGpp effects of RNAP in the original manuscript by looking at the conformational heterogeneity, it could be extremely challenging to trap a dominant ppGpp-bound conformation that would clearly explain the effect of the ppGpp on RNAP in molecular details. Therefore, to address concerns raised by this Reviewer, we removed comparisons and structural explanation for the ppGpp-mediated effects on RNAP conformation.

5. According to the sequences listed in Materials and Methods, the nucleic acid scaffold, which was used to reconstitute the EC or ppGpp-EC contains an artificial bubble of 12 bp not 10 bp as written (p19) – there must be a typo! Recent cryo-EM reconstructions suggest the bubble is likely only 10 and at most 11 bp (template DNA positions -10/-9 to +1) – a larger bubble may have unintended consequences. The authors need to confirm that indeed a 10bp bubble has been used.

For the revised manuscript we conducted new cryo-EM studies with a new scaffold featuring a 10-nt bubble. Even with this new scaffold, we only see a part of the bubble.

6. Irrespective of bubble size, the scaffold contains only 9bp of downstream duplex (position +2 to +10) – this is very short and does not reach the edge of the RNAP pincers (e.g. the jaw domain, which normally interacts with the downstream duplex) – this almost certainly influences the mobility of peripheral domains such as the jaw-domain, the trigger loop insertion domain (SI3 also called bi4), which interacts with the jaw, and the beta2-lobe. As a consequence, I am skeptical about the relevance of increased mobility, which could simply reflect the lack of downstream DNA. Along these lines: why are B-factors of jaw and SI3 lower in the EC compared to the ppGpp-EC if they're supposed to be more mobile in the former (Fig. 3d and Bfactors in provided models)?

We conducted new cryo-EM studies with a scaffold containing longer upstream and downstream DNA. Despite these extensions, we only see clearly 17 out of 53 nts of nontemplate DNA, 24 nt out of 53 nts of template DNA, and 10 out of 16 nts of RNA in both EC and EC-ppGpp. The map for the upstream DNA duplex can be seen at low levels but cannot be unambiguously interpreted.

7. The overall resolution and map-quality of the two reconstructions is superb. However, I have noticed an important difference that is difficult to understand. In contrast to the EC, density for the upstream DNA duplex is completely absent in the ppGpp-EC and the density for the downstream DNA duplex is also weak.

We see identical regions of the scaffold (**new Extended Data Fig. 3a,b**) and similar quality maps in both new EC and EC-ppGpp structures.

More importantly, density for the non-template DNA in the downstream duplex is extremely weak in the ppGpp-EC. Only the template DNA bases for positions +2 to +5 are well resolved.

This is very unusual and I have never seen this in any other EC (swiveled or non-swiveled). This raises a serious concern: Was there a difference in preparing the sample and could the nontemplate DNA have (partially) dissociated due to the short upstream and downstream duplex in the ppGpp-EC (see previous point)? I am concerned because a difference in sample composition would of course not allow to make any structural comparison between the two samples. Also, any structural effect of ppGpp in site 1 would likely be masked by the effect of the absence of the non-template DNA.

Both EC and EC-ppGpp samples were prepared essentially in the same way. In both new EC and EC-ppGpp structures (new **Extended Data Fig. 3a,b**), we can clearly see only 51 nts out of 122. Although we can clearly see only 4 nts of the non-template strand in the transcription bubble, the downstream part of the DNA duplex has a good map, suggesting no dissociation occurred during the sample preparation for the shortest duplex part.

At a minimum, the authors need to provide low-pass filtered maps of the ppGpp-EC to confirm the presence of the non-template DNA – maybe it is more disordered? However, I am not convinced that increased disorder alone will explain the lack of density (especially because it is not the case in the EC, which uses the exact same scaffold). Given the high resolution and large number of particles, focused classification based on the downstream DNA duplex (and possibly upstream duplex) may allow to separate into +/- non-template DNA complexes and should be tried (see first point). Also, this difference in non-template DNA density needs to be discussed and pointed out to the reader and modeling of the non-template DNA is not justified by the maps in my opinion (and reflected in the poor geometry of DNA base pairs).

We thank the Reviewer for this suggestion. The new structures of EC and EC-ppGpp, determined to a higher resolution also have poor maps for a large part of the DNA. In our opinion, both complexes have maps of comparable poor quality and we refrained from modelling DNA into these maps.

Minor comments:

1. It is important to point out to the reader that the scaffold used by the authors would not allow backtracking because of the mismatches in the transcription bubble. Otherwise, one may wonder why biochemical results suggest a tendency for RNAP to backtrack but the structures do not show that.

This is a very good point. We now mention this in the Discussion of the revised manuscript: “As the mismatches in the transcription bubble of the nucleic acid scaffold would not allow backtracking, the EC-ppGpp structure does not capture backtracking-prone conformation.”

As a note on the side: I think labs performing structural biology of RNAP ECs should consider to stop using minimal nucleic acid scaffolds that bias RNAP into a specific translocation state. A complementary non-template DNA would have allowed RNAP to adopt the preferred state and

the resulting reconstructions would reflect the equilibrium between post-, and pre-translocated state and potential backtracked states. We are reaching a point, where these small differences can be disentangled in focused classification approaches (when high-resolution data is available).

We agree with this note. Indeed, longer scaffolds should be more informative in principle, even though our new structures presented in the revised manuscript did not reveal more than a half of the DNA scaffold, arguing that DNA is too flexible in the EC. In an unrelated project, we obtained a mixture of post- and pre-translocation states, which, however, could not be disentangled by focused classification even with the high resolution of the maps.

2. I would recommend to use base pair restraints during refinements for parts of the structure with weaker density. This will help to obtain more realistic geometries for the DNA upstream and downstream duplex.

There was no need to use base pair restrains in the new structures.

3. Is it valid to compare the latent coordinates from 3D variability analysis between two different (the emphasis being on different) datasets to estimate real-space differences in motion as suggested by Fig. 3C? In any case, I think classification into 2 or 3 classes should be straightforward given the size of the dataset and this would allow a comparison of swivel motion between EC and ppGpp-EC.

This is a good point, which does not have a clear answer now. We have attempted the comparisons with a few data sets and, because of differences in the magnitude of changes, ultimately decided not to include the comparisons between 3D variability analysis in the revised manuscript.

4. Fig. 3c – labels are mixed up because it is the ppGpp-EC, which is supposedly narrower than the EC, right?

Yes this was a typo. Original Fig. 3 has been removed from the revised manuscript.

Decision Letter, first revision:

Message: Our ref: NSMB-A45745A-Z

17th Jan 2023

Dear Dr. Nudler,

My name is Dimitris Typas and I am happy to now be the handling editor of your manuscript entitled "Control of Transcription Elongation and DNA Repair by Alarmone ppGpp" (NSMB-A45745A-Z). I sincerely apologise for not reaching you earlier with a decision, resulting from a delay in receiving the final reports from the reviewers due to the end-of-the-year holidays.

Your revised manuscript has now been seen by the original referees and their comments are below. The reviewers find that the paper has improved in revision, and therefore we'll be happy in principle to publish it in Nature Structural & Molecular Biology, pending minor revisions to satisfy the referees' final requests and to comply with our editorial and formatting guidelines.

To facilitate our work at this stage, we would appreciate if you could send us the main text as a word file. Please make sure to copy the NSMB account (cc'ed above).

Sincerely,

Dimitris Typas
Associate Editor
Nature Structural & Molecular Biology
ORCID: 0000-0002-8737-1319

Reviewer #1 (Remarks to the Author):

In the revised manuscript, authors replaced the previous structures (EC and EC-ppGpp) with longer nucleic acid scaffold to address the discrepancies raised. In the new structure, upstream and downstream DNA densities were reported as they appear in the cryo-EM density maps. The position of RNA 3' end in the new structure is in the post-translocation state as previous structures, and they discuss the possibility that because of mismatch transcription bubble, RNAP backtracking is unlikely after ppGpp binding. Further, they updated the metal coordination assignments by ppGpp phosphate groups. As suggested, authors performed new transcription experiments: 1. without ppGpp to confirm that ppGpp strengthens the both P1 and P2 pauses which are susceptible to backtracking, and 2. In the presence of DksA and NusG to see the effect of ppGpp in elongation. They confirmed no effect of DksA/NusG on elongation and same effect of KIDQ mutation as without DksA/NusG. The discussion has been updated and suggestions are incorporated in the revised manuscript. Overall, the revised manuscript improved substantially and authors have addressed my concerns. Therefore, I recommended publication of the manuscript in NSMB.

Reviewer #2 (Remarks to the Author):

The authors' response and revision to my earlier comments are satisfactory. I have no further comments.

Reviewer #3 (Remarks to the Author):

The revised manuscript by Weaver et al. investigates the impact of the alarmone ppGpp on RNA polymerase (RNAP) elongation complexes (EC). The manuscript describes a combination of in vitro, in vivo, and structural approaches to provide insights on the role of ppGpp in transcription-coupled DNA repair.

After raising concerns about inconsistencies in the initial cryo-EM reconstructions, the authors have decided to do two new reconstructions and, consequently, come to different conclusions. The new reconstructions no longer suggest that ppGpp influences the conformational equilibrium of the RNAP EC and aside from a small shift in the omega subunit, the two structures are virtually identical. It is laudable that the authors acknowledged my concerns and decided to determine new EM reconstructions. The high-resolution structures allowed the authors to improve previous models of ppGpp bound to site 1 but unfortunately they do not provide a mechanistic explanation on its role. This is unfortunate and got essentially summarized by the authors with the somewhat euphemistic statement "These results suggest that ppGpp binding to Site 1 likely affects motions of RNAP during the catalytic cycle and that static three-dimensional scaffold-based structures do not capture these effects."

The authors have addressed all my concerns and as a result, I have no further reservations. In the end, it needs to be an editorial decision if the new insights are interesting for a broad enough audience to justify publication in NSMB.

Final Decision Letter:

Message 27th Feb 2023

:

Dear Dr. Nudler,

We are now happy to accept your revised paper "Control of Transcription Elongation and DNA Repair by Alarmone ppGpp" for publication as a Article in Nature Structural & Molecular Biology.

Your paper will be published online soon after we receive proof corrections and will appear in print in the next available issue. You can find out your date of online publication by contacting the production team shortly after sending your proof corrections. Content is published online weekly on Mondays and Thursdays, and the embargo is set at 16:00 London time (GMT)/11:00 am US Eastern time (EST) on the day of publication. Now is the time to inform your Public Relations or Press Office about your paper, as they might be interested in promoting its publication. This will allow them time to prepare an accurate and satisfactory press release. Include your manuscript tracking number (NSMB-A45745B) and our journal name, which they will need when they contact our press office.

About one week before your paper is published online, we shall be distributing a press release to news organizations worldwide, which may very well include details of your work. We are happy for your institution or funding agency to prepare its own press release, but it must mention the embargo date and Nature Structural & Molecular Biology. If you or your Press Office have any enquiries in the meantime, please contact press@nature.com.

If you have not already done so, we strongly recommend that you upload the step-by-step protocols used in this manuscript to the Protocol Exchange. Protocol Exchange is an open online resource that allows researchers to share their detailed experimental know-how. All uploaded protocols are made freely available, assigned DOIs for ease of citation and fully

searchable through nature.com. Protocols can be linked to any publications in which they are used and will be linked to from your article. You can also establish a dedicated page to collect all your lab Protocols. By uploading your Protocols to Protocol Exchange, you are enabling researchers to more readily reproduce or adapt the methodology you use, as well as increasing the visibility of your protocols and papers. Upload your Protocols at www.nature.com/protocolexchange/. Further information can be found at www.nature.com/protocolexchange/about.

Please note that *Nature Structural & Molecular Biology* is a Transformative Journal (TJ). Authors may publish their research with us through the traditional subscription access route or make their paper immediately open access through payment of an article-processing charge (APC). Authors will not be required to make a final decision about access to their article until it has been accepted. [Find out more about Transformative Journals](https://www.springernature.com/gp/open-research/transformative-journals)

Authors may need to take specific actions to achieve [compliance with funder and institutional open access mandates](https://www.springernature.com/gp/open-research/funding/policy-compliance-faqs). If your research is supported by a funder that requires immediate open access (e.g. according to [Plan S principles](https://www.springernature.com/gp/open-research/plan-s-compliance)) then you should select the gold OA route, and we will direct you to the compliant route where possible. For authors selecting the subscription publication route, the journal's standard licensing terms will need to be accepted, including [self-archiving policies](https://www.springernature.com/gp/open-research/policies/journal-policies). Those licensing terms will supersede any other terms that the author or any third party may assert apply to any version of the manuscript.

Sincerely,

Dimitris Typas
Associate Editor
Nature Structural & Molecular Biology

ORCID: 0000-0002-8737-1319
